# Pairwise effects between lipid GWAS genes modulate lipid plasma levels and cellular uptake

Magdalena Zimoń[1,2,8], Yunfeng Huang[3,8], Anthi Trasta[1,2,8], Aliaksandr Halavatyi [4], Jimmy Z. Liu[3], Chia-Yen Chen[3,5], Peter Blattmann [1,2,6], Bernd Klaus[7], Christopher D. Whelan [3], David Sexton [3], Sally John[3], Wolfgang Huber [7], Ellen A. Tsai [3], Rainer Pepperkok[1,2,4,9 ✉] & Heiko Runz [1,3,9 ✉]

Complex traits are characterized by multiple genes and variants acting simultaneously on a phenotype. However, studying the contribution of individual pairs of genes to complex traits has been challenging since human genetics necessitates very large population sizes, while findings from model systems do not always translate to humans. Here, we combine genetics with combinatorial RNAi (coRNAi) to systematically test for pairwise additive effects (AEs) and genetic interactions (GIs) between 30 lipid genome-wide association studies (GWAS) genes. Gene-based burden tests from 240,970 exomes show that in carriers with truncating mutations in both, *APOB* and either *PCSK9* or *LPL* ("human double knock-outs") plasma lipid levels change additively. Genetics and coRNAi identify overlapping AEs for 12 additional gene pairs. Overlapping GIs are observed for *TOMM40/APOE* with *SORT1* and *NCAN*. Our study identifies distinct gene pairs that modulate plasma and cellular lipid levels primarily via AEs and nominates putative drug target pairs for improved lipid-lowering combination therapies.

[1] Molecular Medicine Partnership Unit (MMPU), University of Heidelberg/EMBL, Heidelberg, Germany. [2] Cell Biology and Biophysics Unit, European Molecular Biological Laboratory, Heidelberg, Germany. [3] Translational Biology, Biogen Inc, Cambridge, MA, USA. [4] Advanced Light Microscopy Facility (ALMF), European Molecular Biological Laboratory, Heidelberg, Germany. [5] Psychiatric and Neurodevelopmental Genetics Unit, Mass General Hospital, Boston, MA, USA. [6] Idorsia Pharmaceuticals Ltd, Basel, Switzerland. [7] Genome Biology Unit, European Molecular Biological Laboratory, Heidelberg, Germany. [8]These authors contributed equally: Magdalena Zimoń, Yunfeng Huang, Anthi Trasta. [9]These authors jointly supervised this work: Rainer Pepperkok, Heiko Runz. ✉email: pepperko@embl.de; heiko.runz@gmail.com

Genome-wide association studies (GWAS) have firmly established that a substantial fraction of variation in blood lipid levels and the risk of coronary artery disease (CAD) is heritable. Hundreds of genetic loci have been identified that reach genome-wide significant associations with plasma levels of low-density lipoprotein cholesterol (LDLc), high-density lipoprotein cholesterol (HDLc), triglycerides (TG), total cholesterol (TC) and CAD[1–4]. In rare instances, susceptibility to altered blood lipids can be attributed to mutations in individual genes such as *LDLR*, *PCSK9* or *APOB* that lead to familial forms of disease. For the vast majority of dyslipidemic individuals, however, no single gene mutation can be identified or remain undetected without substantial follow-up. Recent evidence suggests that in the majority of such cases inherited susceptibility is caused by a cumulative effect of numerous common alleles within and across GWAS loci. Individually, such common alleles have only a minor effect, but when summarized in polygenic scores they can modify a phenotype to a similar extent as single high-impact mutations[5], or further magnify the penetrance of individual mutations causing Mendelian disease[6]. The biological mechanisms behind the cumulative effect of risk alleles in different genes remain largely unclear.

While the refined understanding of the polygenic nature of complex disease is starting to show promise for improved risk prediction and treatment decisions[7,8], it has made it increasingly difficult to decide which individual genes could be the most suitable targets for developing new drugs. Drug development is traditionally focused on discrete targets with well-understood biology. For certain diseases, an additive therapeutic benefit has been demonstrated through combination therapies that simultaneously modulate two or more targets at once. For instance, combinations of statins, inhibitors of HMG-CoA-reductase (*HMGCR*), with distinct other cholesterol-lowering medications including *NPC1L1*, *PCSK9* and *APOB* inhibitors have been demonstrated to lower LDLc levels and CAD risk further than statin treatment alone[9,10]. Despite such successes, systematic strategies to predict that joint modulation of drug target pairs in combination therapies will show benefits beyond standard of care have yet to be explored.

Genetic support for a drug target increases the probability that a medicine directed against the respective target will succeed by several fold[11,12]. We thus hypothesized that genetics might also assist in nominating drug target pairs that, when addressed jointly, will have a higher probability to reach a desired therapeutic benefit. A particularly attractive approach to prioritize optimal target pairs would be to leverage synergistic gene-gene interactions, where genetic variants in two disease risk genes induce a phenotype that is more pronounced than what would be expected from each of the variants' individual effects. Genetic interactions (GIs), or epistasis, have been extensively studied in model organisms and cell models with the aim to identify functional relationships among genes and gene products[13–16]. In humans, however, the contribution of GIs to complex traits has been controversial. While there is increasing evidence for modifier genes that modulate Mendelian phenotypes in non-additive manners[17], most of the variance of complex traits appears to be explained by genes acting additively within or between loci (additive effects, or AEs)[18].

Here we systematically test for pairwise genetic effects, both GIs and AEs, that regulate blood lipid levels by studying whether 30 genes prioritized based on known lipid regulatory functions from GWAS loci interact. For this, we use three complementary tools: protein-truncating variants (PTVs) identified through exome sequencing in the UK Biobank; reported GWAS lead SNPs genotyped or imputed in the UK Biobank; and combinatorial RNA interference (coRNAi) screening measuring LDLc uptake into cultured cells. Our combined genetics and functional genomics approach provides evidence that pairwise effects between lipid genes are foundational elements in controlling blood lipid levels and highlights distinct gene pairs as promising targets for lipid-lowering combination therapies.

## Results

**Study outline**. To explore combined effects between genes in GWAS loci and how these impact plasma lipid levels and LDLc uptake into cultured cells, we followed three parallel approaches: first, we extracted protein-truncating variants (PTVs) from whole-exome sequencing data of 302,331 participants of the UK Biobank. Second, we utilized GWAS lead SNPs commonly used to construct polygenic risk scores from the full set of 378,033 unrelated participants of European ancestry in the UK Biobank. And third, we conducted systematic RNAi-based combinatorial knockdown experiments in cells (Fig. 1a). We focussed our analyses on 30 high-confidence candidate genes from 18 genomic regions associated with blood lipid levels or the risk for CAD (Supplementary Data 1). Twenty-eight of these genes had scored as functional regulators of LDLc uptake, cellular levels of free cholesterol, or LDL-receptor (LDLR) mRNA or protein levels in an earlier study where we had functionally analysed 133 genes at 56 lipid and CAD GWAS loci through RNAi-based knockdown experiments[19]. Causality for several of these genes to drive GWAS associations was further supported through systematic colocalization of plasma LDLc GWAS lead SNPs with GTEx liver eQTLs[1] (2 genes), cis-pQTL signals[20] (3 genes) and independently reported biological evidence for lipid-relevant functions (15 genes) (Supplementary Data 2). To identify pairwise effects, we applied four linear regression models (modified from Axelsson et al.[21]) to model the data. For each gene pair, both the additive effects (AEs) (model 3), defined by the sum of effects from each gene or variant individually, as well as the genetic interactions (GIs) (model 4), represented by observed effects different than the expected additive effect, were calculated, with GIs being divided further into either negative (aggravating) or positive (alleviating or suppressive) (Fig. 1a and 'Methods')[22]. Pairwise analyses were conducted for four plasma lipid parameters (LDLc, HDLc, TG, TC) and CAD as available from UK Biobank[23] (see 'Methods').

**PTV burden tests in UK Biobank reveal additive effects for four gene pairs**. We first studied pairwise effects between the 30 lipid candidate genes using predicted high-impact protein-truncating variants (PTVs). PTVs are expected to cause loss-of-function and compared to other types of mutations are rare at the population level due to purifying selection[24,25]. We obtained the quality-controlled exome sequences of 302,331 UK Biobank participants, annotated PTVs using Variant Effect Predictor v96[26] and the LOFTEE plugin[25], and identified 573,369 high confidence PTVs in the canonical transcripts of 19,076 genes. Within the 30 lipid GWAS genes, we detected a total of 983 unique rare PTVs (Supplementary Data 3). For instance, we discovered 41 different PTVs in *LDLR*, 57 in *PCSK9* and 142 in *APOB*. Most PTVs in these three genes co-occured with strongly abnormal plasma LDLc levels in heterozygote carriers, with 45 PTVs annotated as pathogenic or likely pathogenic in ClinVar[27].

Gene-based PTV burden association analyses were conducted in a cohort of 240,970 unrelated UK Biobank participants of European ancestry. Single-gene PTV burden testing identified three genes that were significantly associated (Bonferroni-corrected $p < 0.05$) with both LDLc and TC (*APOB*, *PCSK9*, *LDLR*), one with LDLc (*LPL*), two with HDLc (*LPL, APOB*) and two with TG (*LPL, APOB*), respectively (Supplementary Data 4). Loss-of-function of these genes had already been identified earlier

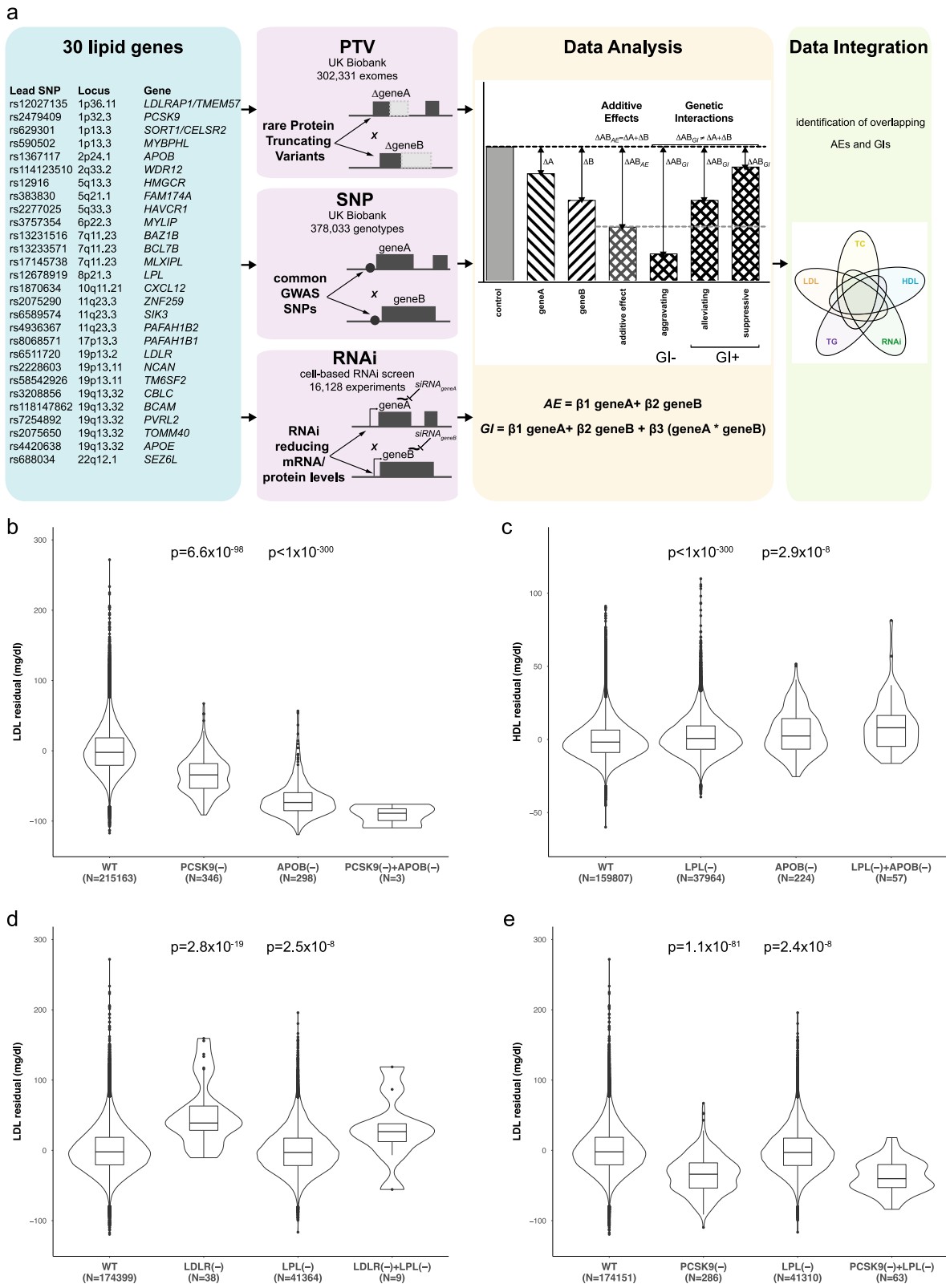

as associated with the respective lipid traits at the population level[2]. Next, we expanded from these single gene PTV burden analyses to study PTV-based pairwise effects, which could be tested for 45 of the 435 theoretically possible gene combinations (Supplementary Data 5 and 'Methods'). Four gene pairs met our stringent criteria to be classified as AEs (*PCSK9-APOB* for LDLc and TC; *LPL-APOB* for HDLc, LDLc and TG; *LDLR-LPL* for

LDLc; and *PCSK9-LPL* for LDLc), reflecting that joint loss-of-function of both genes modulates the respective lipid measures significantly more than if only one of the two genes is truncated (Fig. 1b–e, Supplementary Data 5). For instance, while control individuals without PTVs in either *PCSK9* or *APOB* had average levels of 138.3 mg/dl LDLc, PTVs in *PCSK9* and *APOB* individually reduced mean plasma LDLc by 34.5 mg/dl and

**Fig. 1 PTV burden tests in UK Biobank establish additive effects for *PCSK9-APOB*, *LPL-APOB*, *LDLR-LPL* and *PCSK9-LPL*. a** Workflow of the study. Tests for pairwise genetic effects were conducted with 30 high-confidence candidate genes chosen from 18 GWAS regions associated with blood lipid traits or CAD risk based on colocalization analyses with eQTL/pQTL signals and/or previously reported lipid-regulatory functions (see 'Methods'). Pairwise analyses were conducted from three complementary datasets: protein-truncating variants (PTVs) from exome sequencing in the UK Biobank; lipid/CAD GWAS lead SNPs; and combinatorial RNAi (coRNAi) experiments in cells. Robust linear model fitting was used to identify additive effects (AEs) and genetic interactions (GIs), and genetic and functional data were integrated. **b–d** Gene-based PTV burden AEs, minimum and maximum values of residualized lipid measures as well as the data distribution were shown by violin plots, box in the middle indicates 25 and 75 percentile with a horizontal black line indicating the median **b** between *PSCK9* and *APOB* for LDLc in $n = 215{,}810$ (and TC; Supplementary Fig. 1a), **c** *LPL* and *APOB* for HDLc in $n = 198{,}052$ (and TG, LDLc; Supplementary Fig. 1b), **d** *LDLR* and *LPL* for LDLc in $n = 215{,}810$, and **e** *PCSK9* and *LPL* for LDLc in $n = 215{,}810$ exomes of unrelated UK Biobank participants of European ancestry. *n*, number of carriers. (−), predicted loss-of-function due to PTVs. Shown are *p*-values for single-gene genetic effects derived from robust linear model fit and corrected for multiple comparisons with FDR method (see 'Methods'). For all gene pairs displayed, Interaction $p_{(FDR)}$-values were not significant ($p > 0.01$), consistent with AEs.

69.1 mg/dl relative to individuals without PTVs in these genes, consistent with previous reports[28,29]. However, the three UK Biobank participants ('human double knock-outs') who carried both, *PCSK9* and *APOB* PTVs, showed on average a further reduction in plasma LDLc by 41.2 mg/dl compared to individuals with PTVs in only one of the two genes, and by 91.7 mg/dl compared to individuals with no PTV in either of the two genes (Fig. 1b, Supplementary Fig. 1a), suggesting considerable additional protection from CAD. Similarly, individuals who carried PTVs in both, *LPL* and *APOB*, showed gradually higher HDLc and TG levels than individuals with no PTVs, or PTVs in only one gene (Fig. 1c, Supplementary Fig. 1b). Conversely, for other AEs such as for *LDLR-LPL* and *PCSK9-LPL* for LDLc, *LDLR* and *PCSK9*, respectively, exerted the predominant effect on the respective lipid traits (Fig. 1d, e, Supplementary Fig. 1c, d). Conditioning these gene pairs for the most prevalent *LPL* PTV, p.S447X abrogated the signals, suggesting the AEs with *LPL* were primarily driven through this distinct gain-of-function allele instead of a loss-of-function mechanism (Supplementary Data 6). Our PTV-based burden tests in up to 240,970 exomes did not identify any GIs. This is consistent with the prediction that for rare variant-based burden analyses even larger sample sizes will be necessary to robustly detect GIs in the human population[18,28].

**Pairwise GIs between GWAS lead SNPs modulate plasma lipid levels.** We next tested for pairwise genetic effects using 28 lipid/CAD GWAS lead SNPs representing the 30 loci in 378,033 unrelated individuals of European ancestry in the UK Biobank[23] as proxies for the respective candidate genes. Of a total of 1890 pairwise SNP–SNP effects tested, 142, 41, 78, 140 and three AEs were identified for LDLc, HDLc, TG, TC and CAD, respectively (Fig. 2a–e; Supplementary Data 7). Interestingly, SNP-based analyses also suggested pairwise effects between GWAS loci that deviated from an additive model and were classified as GIs. Specifically, we detected ten GIs for LDLc, one for HDLc, five for TG, and nine for TC (Table 1). No GI was detected for CAD. The strongest driver of putative interactions came from the 19q13.32 *APOE* locus which in our analyses contributed to 19 of the 25 GIs identified across all traits. Twelve GIs were between lead SNPs from within the same GWAS region ('cis-GI'), although only one gene-pair showed strong ($R2 = 0.764$; *NCAN-TM6SF2*) and two weak LD ($R2 > 0.1$; *ZNF259-SIK3*, *ZNF259-PAFAH1B2*) between corresponding lead SNPs. GIs were also identified between loci on different chromosomes ('trans-GIs'), such as between *ZNF259* and *APOE*, or *SORT1/CELSR2* and *TOMM40* for LDLc and TC, or between *LPL* and *ZNF259*, or *LPL* and *SIK3* for TG. Overall, our data support the hypothesis that AEs between GWAS loci are pervasive and individually small, yet if summed up across many loci in polygenic scores modulate complex traits[5]. Conversely, GIs are considerably less prevalent, with the *APOE* locus being a potential contributor to GIs for lipid traits.

**Genetic effects between PTV burden with GWAS lead SNPs and polygenic risk.** Next, we queried for GIs between different types of genetic variation. Pairwise interaction testing between gene-based PTV burden and GWAS lead SNPs identified two GIs, $LDLR_{PTV}$-$APOB_{SNP}$ and $LDLR_{PTV}$-$PCSK9_{SNP}$, both with LDLc as well as TC (Supplementary Data 8). Moreover, 67, 21, 60 and 26 AEs were identified for LDLc, HDLc, TC and TG, respectively. These results are consistent with the types of genetic variation modulating plasma lipids being continuous between high-impact rare and low-impact common alleles[4] and again GIs being substantially less prevalent than AEs. A recent study[6] proposed that the penetrance of Mendelian disease, including FH, can be modulated by genetic effects between the respective mutant gene with common variants (minor allele frequency $>0.01$) of individually small effect size subsumed in polygenic risk scores (PRS). We created PRS for the four lipid species using PRS-CS[29] ('Methods') and tested for genetic effects between PRS and PTV burden for each of the 30 genes. Five AEs were identified (between $APOB_{PTV}$ with PRS for HDLc and TG; $PCSK9_{PTV}$ with PRS for LDLc and TC; and $LPL_{PTV}$ with PRS for LDLc). Of all combinations tested, only PTV burden in *LPL* showed evidence for a GI with the PRS for TG ($p = 1.77 \times 10^{-14}$; beta = −0.03) (Fig. 2f; Supplementary Data 9). The GI remained significant ($p < 2 \times 10^{-16}$, beta = −0.04) in a sensitivity analysis using the p.S447X variant alone instead of *LPL* PTV-burden, proposing an alleviating GI of p.S447X variant on TG-PRS. These results are consistent with the hypothesis that a high polygenic risk for elevated TG can be mitigated by a concomitant gain-of-function mutation in *LPL*.

**RNAi identifies pairwise gene interactions modulating LDLc uptake.** To gain insights into the functional consequences of genetic effects, we complemented our genetic analyses with systematic experiments in cells using combinatorial RNAi (coRNAi) (Fig. 3a and 'Methods'). We applied solid-phase reverse transfection to simultaneously knockdown candidate gene pairs in cultured HeLa cells, which we have previously shown to reliably reflect various aspects of LDLc biology and lipid homoeostasis[19,30,31]. Each of the 30 lipid genes was profiled with a single siRNA that had previously been validated to significantly enhance or reduce cellular uptake of fluorescently-labelled LDL (DiI-LDL) or free cellular cholesterol levels, and/or to efficiently downregulate mRNA or protein levels of its respective target gene (Supplementary Data 2)[19]. The impact of both, single and combinatorial gene knockdown on LDLc uptake per cell was measured and quantified from high-content microscopy images using automated image analysis routines as described (Supplementary Fig. 2)[30,31]. All pairwise knockdown combinations between the 30 lipid genes (435 gene pairs) were assayed in a total of 16,128 experiments (Fig. 3b). Each combination was tested in at least seven biological replicates. In order to identify genetic

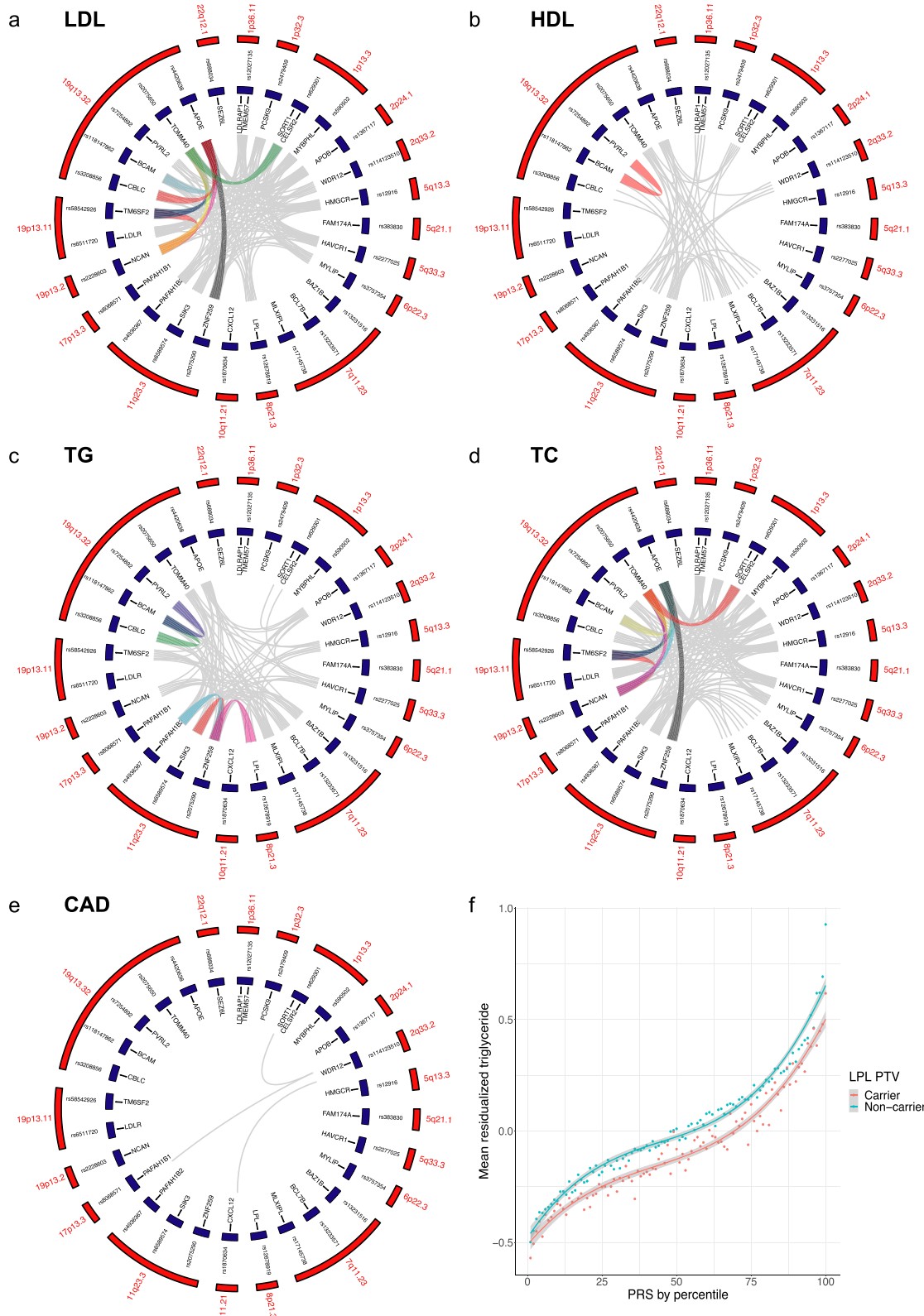

**Fig. 2 Pairwise genetic effects between lipid and CAD GWAS lead SNPs in 387,033 UK Biobank participants. a–e** Circos plots showing AEs (grey) and GIs (coloured) between GWAS lead SNPs (blue) at the 28 selected lipid/CAD loci (red) for the four tested lipid species and CAD. **f** Tests for GIs between polygenic risk scores (PRS) for the four lipid species and PTV burden for each of the 30 lipid genes identified a GI between PTV burden in *LPL* and the PRS for TG. PRS distribution (mean ± SD) for *LPL*-PTV carriers (pink) and non-carriers (blue) are plotted against mean normalized residual TG levels. Each dot reflects mean TG levels at a respective percentile.

**Table 1 Genetic interactions from pairwise PTV-burden and GWAS lead SNP-based GI testing in UK Biobank.**

| Trait | Gene 1 | Gene 2 | BIC_Best_Model | Lowest ΔBIC | incl. 19q13.32 | cis GI | trans GI | MAF (SNP1) | N_PTV1 carriers | MAF (SNP2) | N_PTV2 carriers | SNP1_SNP2 p-value(FDR) |
|---|---|---|---|---|---|---|---|---|---|---|---|---|
| LDLc SNP-SNP GIs: | | | | | | | | | | | | |
| LDLc | NCAN (rs2228603) | TM6SF2 (rs58542926) | 4 | 85.34 | | + | | 0.076 | | 0.076 | | 1.68E−25 |
| LDLc | TM6SF2 (rs58542926) | APOE (rs4420638) | 4 | 63.34 | + | (+) | | 0.191 | | 0.191 | | 5.17E−19 |
| LDLc | TM6SF2 (rs58542926) | TOMM40 (rs2075650) | 4 | 41.95 | + | (+) | | 0.147 | | 0.147 | | 1.11E−13 |
| LDLc | BCAM (rs1181147862) | APOE (rs4420638) | 4 | 34.25 | + | + | | 0.046 | | 0.191 | | 2.45E−13 |
| LDLc | NCAN (rs2228603) | APOE (rs4420638) | 4 | 32.61 | + | (+) | | 0.076 | | 0.191 | | 3.38E−12 |
| LDLc | NCAN (rs2228603) | TOMM40 (rs2075650) | 4 | 30.02 | + | (+) | | 0.076 | | 0.147 | | 1.41E−11 |
| LDLc | BCAM (rs1181147862) | TOMM40 (rs2075650) | 4 | 28.81 | + | + | | 0.046 | | 0.147 | | 5.00E−11 |
| LDLc | ZNF259 (rs2075290) | APOE (rs4420638) | 4 | 4.68 | + | | + | 0.068 | | 0.191 | | 6.56E−05 |
| LDLc | CBLC (rs3208856) | APOE (rs4420638) | 4 | 3.34 | + | + | | 0.036 | | 0.191 | | 6.20E−04 |
| LDLc | SORT1/CELSR2 (rs629301) | TOMM40 (rs2075650) | 4 | 0.57 | + | | + | 0.222 | | 0.147 | | 1.11E−03 |
| HDLc SNP-SNP GIs: | | | | | | | | | | | | |
| HDLc | BCAM (rs1181147862) | PVRL2 (rs7254892) | 4 | 1.77 | | + | | 0.046 | | 0.031 | | 3.26E−04 |
| TG SNP-SNP GIs: | | | | | | | | | | | | |
| TG | ZNF259 (rs2075290) | SIK3 (rs6589574) | 4 | 31.81 | | + | | 0.068 | | 0.084 | | 1.29E−09 |
| TG | BCAM (rs1181147862) | PVRL2 (rs7254892) | 4 | 21.41 | + | + | | 0.046 | | 0.031 | | 7.44E−08 |
| TG | CBLC (rs3208856) | BCAM (rs1181147862) | 4 | 20.45 | + | + | | 0.036 | | 0.046 | | 3.72E−07 |
| TG | ZNF259 (rs2075290) | PAFAH1B2 (rs4936367) | 4 | 17.82 | | + | | 0.068 | | 0.1 | | 5.57E−07 |
| TG | LPL (rs12678919) | ZNF259 (rs2075290) | 4 | 13.81 | | | + | 0.098 | | 0.068 | | 6.45E−04 |
| TC SNP-SNP GIs: | | | | | | | | | | | | |
| TC | NCAN (rs2228603) | TM6SF2 (rs58542926) | 4 | 74.81 | | + | | 0.076 | | 0.076 | | 1.35E−21 |
| TC | TM6SF2 (rs58542926) | APOE (rs4420638) | 4 | 53.33 | + | (+) | | 0.076 | | 0.191 | | 3.95E−16 |
| TC | TM6SF2 (rs58542926) | TOMM40 (rs2075650) | 4 | 38.17 | + | (+) | | 0.076 | | 0.147 | | 1.92E−12 |
| TC | NCAN (rs2228603) | APOE (rs4420638) | 4 | 30.93 | + | (+) | | 0.076 | | 0.191 | | 6.22E−12 |
| TC | NCAN (rs2228603) | TOMM40 (rs2075650) | 4 | 28.59 | + | (+) | | 0.076 | | 0.147 | | 5.07E−11 |
| TC | BCAM (rs1181147862) | TOMM40 (rs2075650) | 4 | 11.24 | + | + | | 0.046 | | 0.147 | | 1.68E−07 |
| TC | BCAM (rs1181147862) | APOE (rs4420638) | 4 | 9.05 | + | + | | 0.046 | | 0.191 | | 2.22E−07 |
| TC | ZNF259 (rs2075290) | APOE (rs4420638) | 4 | 2.40 | + | | + | 0.147 | | 0.191 | | 1.24E−04 |
| TC | SORT1/CELSR2 (rs629301) | TOMM40 (rs2075650) | 4 | 0.31 | + | | + | 0.222 | | 0.147 | | 1.56E−03 |
| LDLc PTV-SNP GIs: | | | | | | | | | | | | |
| LDLc | LDLR | PCSK9 (rs2479409) | 4 | 11.29 | | | + | 47 | | 0.349 | | 2.01E−05 |
| LDLc | LDLR | APOB (rs1367117) | 4 | 3.51 | | | + | 47 | | 0.339 | | 1.26E−03 |
| TC PTV-SNP GIs: | | | | | | | | | | | | |
| TC | LDLR | PCSK9 (rs2479409) | 4 | 10.03 | | | + | 47 | | 0.349 | | 8.72E−05 |
| TC | LDLR | APOB (rs1367117) | 4 | 8.37 | | | + | 47 | | 0.339 | | 1.81E−04 |

Genetic interactions (GIs) identified through GWAS lead SNP- and PTV-SNP-based GI analyses in the UK Biobank as described in 'Methods'. A lowest ΔBIC of '4' indicates interaction model is most compatible with a genetic interaction. BIC, Bayesian Information Criterion. MAF (minor allele frequency) estimates and numbers of rare protein-truncating variant (PTV) carriers are based on genotypes from 387,033 and exomes, respectively, from 240,970 unrelated UK Biobank participants of European ancestry. (+) indicates possible cis-effects of rs4420638 in APOE on neighbouring genes on Chr.19q13.32. Trans GI indicates genes contributing to pairwise naGIs are located on different chromosomes. Shown are p-values for SNP-SNP and PTV-SNP interaction effects derived from robust linear model fit and corrected for multiple comparisons with FDR method (see 'Methods').

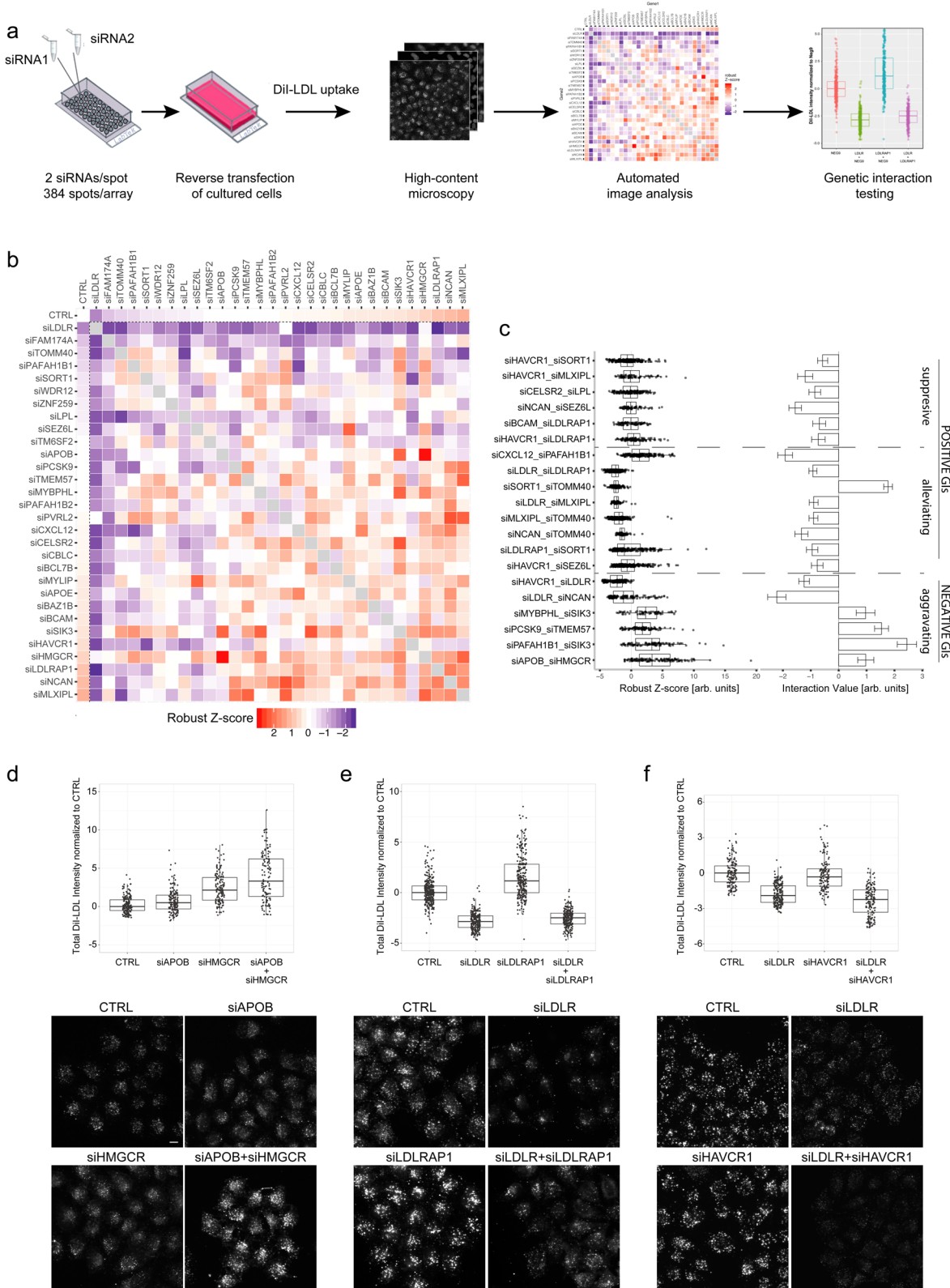

interactions and additive effects, we conducted, analogously to the genetic analyses, robust linear regression model fitting. Initially, we used Schwarz's BIC model estimation to determine potential GIs and AEs. Then we calculated $p$-values from $t$-values of the linear regression model term for single and double knockdowns (see 'Methods'). We corrected these $p$-values for multiple comparison testing with FDR. This identified 37 gene pairs, for which

the interaction coefficient had a $p$-value$_{(FDR)} < 0.005$, proposing them as GIs that differentially impact cellular LDLc uptake (Supplementary Data 10). The corresponding gene pairs were brought forward to independent liquid-phase based coRNAi replication experiments which validated 20 of these GIs (Table 2, Supplementary Data 11, Supplementary Fig. 3). Of the 20 validated GIs identified through coRNAi, six were classified as

**Fig. 3 Combinatorial RNAi identifies pairwise GIs modulating cellular LDLc uptake. a** The coRNAi screen workflow. Customized cell microarrays were generated by pairwise spotting of siRNAs against two different candidate genes on 384 spots/array for solid-phase reverse siRNA transfection of cultured HeLa cells. Integrated fluorescence intensities of internalized fluorescently-labelled LDL (DiI-LDL) for each cell individually were quantified by automated image analysis. Averaged signal intensities per gene pair were tested for genetic effects in multiple replica experiments per array. Effects suggested in the coRNAi screen as non-additive were subsequently validated in customized experiments using fluid-phase transfection. **b** Heatmap visualizing median robust Z-score distribution upon coRNAi of 435 gene pairs assessed for their impact on cellular LDLc uptake (Supplementary Data 10). Red, increase. Blue, decrease. CTRL (top row and first column) reflects the relative impact on LDLc uptake when candidate genes were silenced individually (siRNA$_{geneAorB}$ + CTRL). **c** 20 gene pairs validated as either positive alleviating and suppressive or negative aggravating GIs on cellular LDLc uptake in independent replica experiments. Best estimate of Interaction Value (**c**-right graph) depicts the directionality and difference of the combined effect versus single knockdown effects. Error bars represent ± standard error of the estimate for each interacting pair. **d**–**f** Selected examples of single gene (siRNA$_{geneA}$ + CTRL) and gene pair (siRNA$_{geneA}$ + siRNA$_{geneB}$) siRNA knockdown effects on relative fluorescently-labelled LDLc (DiI-LDL) cellular uptake. CTRL, negative control siRNA. Dots in boxplots ((**c**—left)–(**f**)) are robust Z-score values calculated for integrated DiI-fluorescence intensities of cells averaged per image ('Methods'), showing the data distribution with minimum and maximum values. Boxplots represent values between 25th and 75th percentile, horizontal black line—median robust-Z-score for total $n = 625, 1063, 674$ images for (**d**), (**e**), (**f**), respectively, for (**c**—left graph) number of images is detailed in Supplementary Data 11. All images are originating from minimum three independent biological replicas. Whiskers indicate largest value within 1.5 times interquartile range above 75th percentile. All validated GIs ((**c**)–(**f**)) have $p_{(FDR)} < 0.01$, $p$-values were derived from robust linear model fit and corrected for multiple comparisons with FDR method (see 'Methods'). Scale bar = 10 μm.

negative (GI−)—aggravating (or synergistic), i.e., simultaneous knockdown of both genes magnified the effect size beyond expectations for an additive effect; and fourteen GIs were categorized as positive (GI+). Positive GIs were further subdivided into: seven alleviating GIs (i.e., the joint effect was approximately equal to the most severe phenotype) and seven suppressive GIs (i.e., the joint knockdown was less severe ('healthier') than the most severe of the phenotypes) (Fig. 3c)[22]. For instance, simultaneous knockdown of *HMGCR* and *APOB* enhanced cellular LDLc uptake beyond a mere additive effect expected from knockdown of either of the two genes, proposing an aggravating GI (Fig. 3d), that can most likely be explained by a higher capacity of cells to bind and internalize LDLc via increased availability of LDL receptor at the cell surface (Supplementary Fig. 4). Conversely, knockdown of *LDLR* strongly inhibited, whereas knockdown of *LDLRAP1* increased cellular LDLc uptake under our experimental conditions, an effect that could not be explained by elevated LDLR levels at the plasma membrane (Supplementary Fig. 5) or would have been anticipated from LDLRAP1's role in FH[32]. When silencing *LDLR* and *LDLRAP1* jointly, the reduction of LDLc uptake was less attenuated than expected under an additive model, suggesting a positive alleviating GI (Fig. 3e). Interestingly, reduction of LDLc uptake upon knockdown of *LDLR* was magnified when *LDLR* was jointly silenced with *HAVCR1*, a suggested LDLc scavenger receptor that might contribute to maintenance of the potential of LDLR-depleted cells to internalize LDLc[33] (Fig. 3f). Noteworthy, among the remaining validated coRNAi GIs, simultaneous silencing of *PCSK9* and *TMEM57*, *MYBPHL* and *SIK3* as well as of *SIK3* and *PAFAH1B1* increased cellular LDLc uptake to a similar extent as the simultaneous knockdown of *HMGCR* and *APOB*, although silencing of these genes individually had a significant, yet only modest impact on cellular LDLc uptake. GIs such as these that stimulate LDLc internalization beyond expectation might harbour particular potential as starting points for improved LDLc lowering therapies. In summary, coRNAi identified AEs and GIs between established lipid-regulatory genes, but also proposed combinations of less characterized genes as potentially important factors in maintaining cellular lipid levels.

**Integrated analysis highlights overlapping genetic effects.** In order to assess whether AEs and GIs identified through either PTV-based gene-burden tests, GWAS lead SNPs, or cell-based coRNAi overlapped, we integrated results from the three approaches (Fig. 4). *HMGCR-LDLR* and *LDLR-SIK3* showed AEs

both in coRNAi and PTV-SNP analyses for LDLc and TC (Fig. 4a). There were no overlapping GIs between coRNAi and PTV-burden analyses, although several gene pairs that were classified by coRNAi as GIs were classified by genetics as AEs (Fig. 4b). Eight of the 13 gene pairs nominated by coRNAi as AEs also scored as AEs in SNP-based interaction testing of which five involved *HMGCR* and three *LDLRAP1* (Fig. 4c). Two gene pairs, *SORT1-TOMM40* and *NCAN-TOMM40*, scored as GIs both in the SNP-based as well as the coRNAi-based interaction testing (Fig. 4d). Among these, *TOMM40* exerted an alleviating positive GI effect in either gene pair (Fig. 4e) that could not be explained by an off-target effect of *TOMM40* siRNAs on *APOE* as an adjacent gene in the 19q13.32 GWAS locus (Supplementary Fig. 6). In conclusion, integrating genetic with functional data validated 10 proposed AEs and further substantiated a role of the *APOE* locus, presumably mediated through *TOMM40*, as contributing to genetic interactions.

## Discussion

Here, we apply whole-exome sequencing, genotyping and coRNAi to systematically test for pairwise genetic effects between 30 lipid-regulatory genes at lipid and CAD GWAS loci. Pairwise genetic effects, which encompass either additive effects or non-additive genetic interactions, are considered to be central constituents of biological pathways and complex traits, contributors to human disease, and promising starting points for therapy development[13,17]. Mapping especially GIs, however, has been challenging. GI studies require very large population sizes in order to obtain sufficient statistical power, so that the large number of potential interactions to be evaluated quickly leads to a prohibitive number of statistical tests[34]. Together with most GI studies to date being limited to just a single data type, the relative contribution of GIs to variation in human complex traits has been controversial, and the relevance of epistasis potentially overestimated[18].

In our study, we have tried to overcome several of these challenges through a systematic approach to testing genetic effects that integrates genetic with functional data and relies on the UK Biobank, a population cohort linking genetic with phenotype data at an unprecedented scale[23]. To protect against statistical penalties from multiple hypothesis testing we focused on pairwise interaction analyses between 30 candidate genes nominated through GWAS that functional or genetic follow-up studies have proposed as likely causal to confer associations with lipid traits or CAD[19]. We assessed these genes for genetic effects across the

**Table 2 Pairwise GIs identified and validated through coRNAi to impact LDLc uptake into cells.**

| GI Gene Pair | | Interaction type | Primary RNAi screen | | | Validation RNAi screen | | |
|---|---|---|---|---|---|---|---|---|
| Gene1 | Gene2 | | Robust Z-score | Interaction Value | p-val(FDR) | Robust Z-score | Interaction Value | p-val(FDR) |
| APOB | HMGCR | NEGATIVE / Aggravating | 2.8 | 2.18 | 1.47E−03 | 3.33 | 0.97 | 1.27E−03 |
| HAVCR1 | LDLR | | −2.18 | −1.32 | 5.99E−03 | −2.24 | −1.24 | 2.72E−10 |
| LDLR | NCAN | | −1.86 | −1.28 | 3.56E−03 | −1.23 | −2.22 | 3.39E−11 |
| MYBPHL | SIK3 | | 2.17 | 1.59 | 8.55E−04 | 2.3 | 0.96 | 5.85E−03 |
| PAFAH1B1 | SIK3 | | 1.62 | 1.79 | 4.50E−04 | 3.3 | 2.44 | 7.06E−12 |
| PCSK9 | TMEM57 | | 1.46 | 1.76 | 5.27E−04 | 1.79 | 1.53 | 1.89E−09 |
| HAVCR1 | SEZ6L | | −1.19 | −1.91 | 8.19E−05 | −0.67 | −0.77 | 3.82E−04 |
| LDLR | LDLRAP1 | POSITIVE / Alleviating | −2.44 | −1.86 | 6.66E−06 | −2.49 | −0.92 | 5.18E−10 |
| LDLR | MLXIPL | | −2.03 | −1.49 | 4.14E−04 | −2.31 | −0.90 | 3.79E−08 |
| LDLRAP1 | SORT1 | | −1.11 | −1.65 | 7.94E−04 | −1.09 | −0.95 | 6.18E−06 |
| MLXIPL | TOMM40 | | −2.14 | −2.67 | 5.38E−08 | −1.94 | −0.90 | 3.79E−08 |
| NCAN* | TOMM40* | | −1.4 | −1.56 | 3.23E−03 | −1.49 | −1.34 | 3.45E−08 |
| SORT1* | TOMM40* | | 0.77 | 1.84 | 9.95E−04 | −2.48 | 1.77 | <1.00E−15 |
| BCAM | LDLRAP1 | Suppressive | −0.4 | −1.83 | 3.31E−05 | −0.05 | −0.7 | 4.03E−03 |
| CELSR2 | LPL | | −1.4 | −1.43 | 3.24E−03 | −0.11 | −0.86 | 8.11E−05 |
| CXCL12 | PAFAH1B1 | | −2.17 | −1.78 | 1.13E−04 | 1.32 | −1.92 | 1.77E−12 |
| HAVCR1 | LDLRAP1 | | −0.56 | −2.16 | 6.87E−07 | 0.39 | −0.73 | 4.12E−03 |
| HAVCR1 | MLXIPL | | −0.63 | −2.31 | 1.38E−07 | −0.12 | −1.2 | 2.07E−05 |
| HAVCR1 | SORT1 | | −2.09 | −2.24 | 2.26E−05 | −0.63 | −0.58 | 2.39E−03 |
| NCAN | SEZ6L | | −0.59 | −1.6 | 1.18E−03 | −0.08 | −1.56 | 1.26E−11 |

Gene pairs identified and independently validated by combinatorial RNAi (coRNAi) as impacting the uptake of fluorescently-labelled LDLc into cells in a non-additive manner. Both, BIC, Bayesian Information Criterion and RLMF, Robust Linear Model Fitting were applied for analysis of coRNAi-based GI-testing as described in 'Methods'. Shown p-value derived from robust linear model fit and corrected for multiple comparisons with FDR method (see 'Methods'). *, gene pairs that both, genetic and coRNAi GI-testing proposed as GIs.

allelic spectrum, from rare PTVs ascertained from the exomes of more than 240,000 individuals, to common GWAS lead SNPs. Genetic GI-testing was complemented by functionally knocking down gene pairs with siRNAs and determining the consequence on LDLc internalization into cells.

Several of the genetic effects identified in our study can be expected to be high potential starting points for the development of advanced lipid-lowering combination therapies. Lowering LDLc with statins is the first-line pharmacological strategy to treat or prevent CAD and ischaemic heart disease as its clinical manifestation. However, many patients do not reach their recommended goals of LDLc lowering through statins alone, or they are intolerant against statins. For these, combination therapies have become available that aim to lower atherogenic lipid levels further. A motivation for this is that every 1 mmol/l (39 mg/dl) reduction in blood LDLc is associated with a 19% reduction in coronary mortality and a 21% reduction in major vascular events, supporting that, at least for secondary prevention, the lower blood LDLc levels, the better[35]. Among the options to successfully lower atherogenic blood lipids are therapeutics against drug targets that when mutated cause familial hypercholesterolemia (FH), such as NPC1L1, the target of ezetimibe, or PCSK9. Genetic analyses in extreme phenotypes have identified a small number of individuals with concomitant mutations in two distinct FH genes, such as LDLR and APOB[36,37], LDLR and LDLRAP1[38,39] or APOB and PCSK9[40]. However, due to the rarity of highly penetrant FH mutations such findings have thus far remained limited to individual families. Conversely, on a population level, a previous GI analysis based on common alleles from ~24,000 individuals ascertained for lipid traits reported 14 replicated GIs between lipid GWAS loci, most notably, like in our study, with SNPs at the APOE locus being a key contributor[41]. Additional support for the relevance of genetic effects for modulating lipid traits comes from a study that includes a subset of the UK Biobank exomes analysed here and proposes an interplay of genetic variation across the allelic spectrum[6]. Notably, that study reports that carriers of monogenic CAD risk variants show an up to 12.6-fold higher risk to manifest disease if they are in the highest quintile of the polygenic risk distribution.

Our analyses here propose distinct gene pairs that modulate plasma and cellular lipid levels via additive and non-additive effects. Among others, we identify pairwise effects for several prominent cardiovascular risk genes that individually are established targets for lipid-lowering drugs. For instance, coRNAi proposed a synergistic, aggravating GI between HMGCR, the rate-limiting enzyme during cholesterol biosynthesis and target of statins, and APOB encoding apolipoprotein B, a critical constituent of LDLc particles. Consistent with the known biological functions of these genes, joint knockdown increased levels of functional LDLc receptor on the cell surface and stimulated internalization of exogenous, fluorescently-labelled LDLc. This observation is well in line with results from clinical trials showing that in patients with Familial Hypercholesterolemia and other hyperlipidemias a combination of statins with an antisense inhibitor of apolipoprotein B (mipomersen) efficiently reduces plasma LDLc levels more strongly than high-intensity statin treatment alone[42–45]. Importantly, the AE identified from UK Biobank participants carrying PTVs in both APOB and PCSK9 suggests that similarly beneficial effects can be expected when APOB antisense therapies are applied in combination with PCSK9 inhibitors. Recently, inclisiran, an siRNA targeting PCSK9 in individuals on maximally tolerated statin doses[46] led to a persistent, highly significant lowering of LDLc in treated individuals relative to placebo in a phase 3 study[47], introducing siRNAs as an attractive therapeutic modality for lipid-lowering therapies. Our results strongly propose that, on a population level,

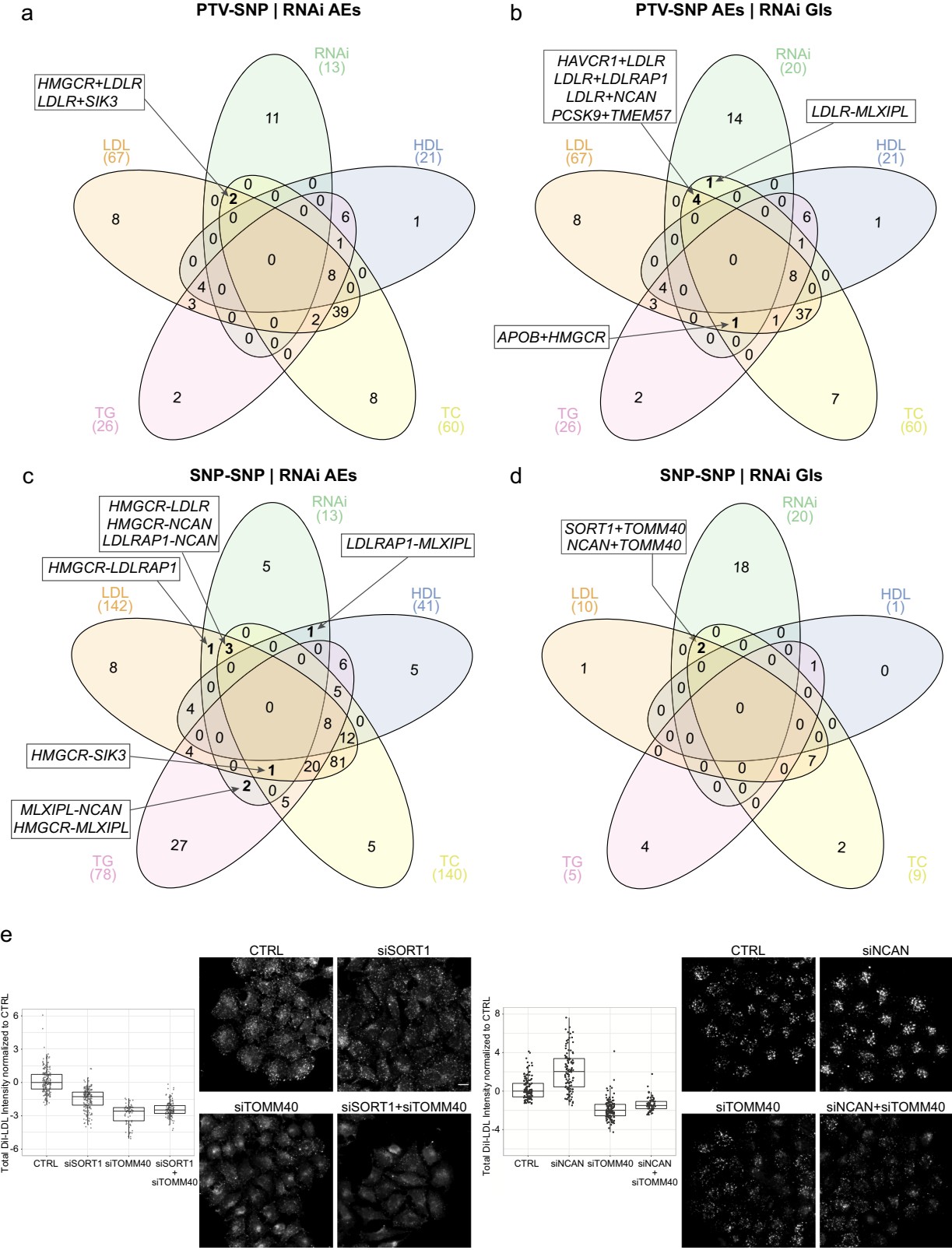

combination therapies inhibiting both *PCSK9* and *APOB* may lower LDLc levels and CAD risk even more substantially than drugs targeting only one of the two genes.

*APOB* PTV burden was associated not only with LDLc and TC, but also HDLc and TG, and our analyses propose that joint disruption of *APOB* together with gain-of *LPL* function reduces TG and increases HDLc, most likely in an additive manner. *LPL*

encodes for lipoprotein lipase which hydrolyzes TG from apoli-poprotein B containing lipoproteins, releasing fatty acids[48]. PTV burden in *LPL* was dominated by the stop-gain variant p.S447X (c.1421 G>C; rs328) which in our exome-sequenced UK Biobank sub-cohort showed an allele frequency of 9.95%. This variant is known to cause gain-of LPL activity leading to a 0.8-fold reduced risk for ischaemic heart disease[49], an effect that is likely to be

**Fig. 4 Integrative analysis identifies pairwise effects supported by both, genetic and functional data.** Overlap of genetic effects identified through genetic analyses and coRNAi. Highlighted are gene pairs identified through either **a**, **b** PTV-SNP or **c**, **d** SNP-SNP testing for which pairwise siRNA-knockdown showed corresponding effects on cellular LDLc uptake. **e** *TOMM40* as an example for which, consistent with SNP-SNP analyses, siRNA knockdown revealed *alleviating positive* GIs when jointly silenced with *SORT1* (left panel) or *NCAN* (right panel). Values on the graphs reflect robust Z-scores values calculated for total intensity of DiI-LDL per cell averaged per image (see 'Methods'). Dots on boxplots represent robust Z-score values calculated for integrated DiI fluorescence intensities of cells averaged per image ('Methods'), showing the data distribution including minimum and maximum values. Boxplots represent values between 25th and 75th percentile, horizontal black line indicates median robust-Z-score for total $n = 588$ and 438 images for (SORT1-TOMM40) and (NCAN-TOMM40), respectively, across three independent biological replicas, whiskers indicate the largest value within 1.5 times interquartile range above 75th percentile. All p-values were derived from robust linear model fit and were corrected for multiple comparisons with FDR method (see 'Methods'); $p_{(FDR)} < 1e^{-15}$, $p_{(FDR)} = 3.45e^{-08}$ for SORT1-TOMM40 and NCAN-TOMM40, respectively. Scale bar = 10 μm.

further enhanced by concomitant reduction of apolipoprotein B. The p.S447X allele was also the main driver behind the only GI detected between PTV burden and polygenic risk for plasma lipids and conferred that in *LPL* PTV-carriers polygenic risk for TG is reduced, with assumed non-additive effects being the most pronounced in the upper percentile range of the PRS distribution.

A prominent driver of GIs in both our SNP- and coRNAi-based analyses was the 19q13.32 locus which includes *APOE* and apart from plasma lipids and CAD is associated with Alzheimer's disease, longevity and macular degeneration among others[20]. Interestingly, our findings indicate that genes other than *APOE* at this locus might contribute to lipid GIs, which is consistent with our earlier findings that knockdown of several genes at this locus independently modulate cellular LDLc uptake[19]. For instance, both SNP-based GI testing and coRNAi suggested alleviating GIs for *TOMM40* with *SORT1* and *NCAN*, respectively. Variants in *TOMM40* have been hypothesized to modify onset of Alzheimer's disease independently of and in conjunction with *APOE*[50]. Our analyses suggest *TOMM40* might exert similar modifying effects on lipid phenotypes and CAD risk, which will need to be clarified in future studies. Such studies will also need to critically assess whether GI signals involving the APOE locus might be inflated due to its large effect sizes[51]. Importantly, our integrated analysis revealed AEs between *MLXIPL*, *NCAN* and *SIK3* with *HMGCR*, nominating these poorly characterized genes to be explored as potentially attractive new targets for lipid-lowering therapies on top of statins.

Consistent with previous assumptions[18], our results show that for regulating plasma lipid levels, additive effects between gene or variant pairs are common, while non-additive epistasis is rare. Indeed, despite a sample size of over 240,000 exomes, our gene-based PTV burden GI analyses did not find evidence for pairwise GIs between lipid genes disrupted by PTVs. Further increasing sample sizes might help uncover non-additive effects, however, at least for lipid traits and based on the analysed candidate gene set, the contribution of GIs to the overall variance appears to be small. This is consistent with the existence of evolutionary mechanisms that suppress epistatic interactions[13]. Since pairwise GIs can be expected to be identified the most easily for genes that are disrupted sufficiently frequently in a population by PTVs of large-enough effect size, sequencing of consanguineous or bottlenecked populations might improve the detection rate of GIs[24,52]. Interestingly, as observed also here, GIs seem to be more easily detectable in cell and animal models, for instance through synthetic lethality mapping[14].

Integration of population-scale genetics and functional coRNAi screening results yielded a total of ten AEs and two putative GIs that influence plasma and cellular lipid levels. Such validation via two systematic approaches substantially increases the confidence for committing to time and resource-intense follow-up analyses of such findings, e.g., when exploring the suitability of a gene pair to be jointly targeted in combination therapies. Interestingly, a

significant number of GIs identified through genetics and coRNAi in our study do not yet overlap. This may be explained by several reasons: First, our functional analyses were limited to measuring LDLc uptake into cells, which reflects a relevant, yet only a partial aspect of the many possible mechanisms by which a gene can modulate plasma lipid levels. Second, siRNA-based gene knockdown captures acute and rather severe functional effects, which may differ from the chronic and often compensated consequences upon lifelong modulation of a gene's function through genetic variation. Third, despite the large number of samples used for genetics-based GI testing, the number of informative high-impact variants in the human germline may still be too discrete to comprehensively identify GIs. Conversely, the large numbers of individually low-impact GWAS lead SNPs may not unambiguously be mapped to a respective gene, so that observed associations may stem from a causal variant from another gene in LD. Regardless, the availability and rapid development of advanced high-throughput microscopy technology joint with the constantly increasing cohort sizes for genetic analyses will allow up-scaling of the approach taken here in future studies and with a high probability identify and validate further genetic effects.

In conclusion, our study introduces a strategy to link large-scale genetic data from a population biobank with quantitative, cell-based coRNAi to map pairwise genetic effects that affect blood lipid levels and CAD, an approach that can be applied to other diseases and complex traits. Our analyses support that mechanisms exist through which pairs of genes help maintain blood lipid homoeostasis in additive and non-additive manners. CAD and ischaemic heart disease remain a substantial global health burden, and doubling-down on lowering atherogenic plasma lipids remains one of the most promising therapeutic approaches. With the encouraging results from recent gene- and antisense-based clinical trials for CAD, our results may help prioritize drug target pairs for the development of lipid-lowering combination therapies rooted in human genetics.

## Methods

**Gene selection**. We chose to study 30 candidate genes from 18 loci reported as associated through common-variant genome-wide association studies (GWAS) with plasma lipid levels and the risk for CAD. Twenty-eight of these genes had been identified and validated as functional regulators of LDLc uptake and/or cholesterol levels into cells in a previous RNAi-screen analysing a total of 133 genes in 56 lipid and CAD GWAS loci[19] (Supplementary Data 1). Common-variant association signals and published biological evidence for potential roles in lipid regulation were updated for all 30 candidate genes based on the recent literature (e.g., refs. [1–3,53]) and queries using the PhenoScanner platform[54] (http://www.phenoscanner.medschl.cam.ac.uk/). Twenty-eight genes were validated to reside within loci that are associated at genome-wide significance ($p < 5e−8$) with plasma lipid levels or CAD. For all lipid loci analysed we selected the more recently published sentinel SNP. When a respective locus was associated with multiple lipid phenotypes, the SNP with the lowest reported p-value association with LDLc was chosen to be the lead SNP. SNPs near *FAM174A* (rs383830) and *SEZ6L* (rs688034) had originally been reported as associated with CAD[55], but failed to replicate at genome-wide significance in more recent meta-GWAS. However, since knockdown

of both genes had scored as significantly impacting lipid parameters in cells[19] the two genes were maintained for this current study.

**Colocalization analysis**. Colocalization analysis was performed between the 28 GWAS lead SNPs using summary statistics from the 2013 Global Lipid Genetics Consortium GWAS[1] (http://csg.sph.umich.edu/willer/public/lipids2013/) and the GTEx liver cis-eQTL dataset ($N = 153$)[56]. When a respective locus was associated with multiple lipid phenotypes, the SNP with the lowest reported $p$-value association with LDLc was chosen to be the lead SNP. There was no GTEx liver expression data for four genes (*APOE, MYBPHL, NCAN, SEZ6L*), therefore there were no cis-eQTL for these genes to colocalize with. Colocalization analysis was conducted following the methods in Giambartolomei et al., [57] using the R 'coloc' package on a ±500 kb window around each lead SNP against SNP-to-expression data of all neighbouring genes within that locus. Positive colocalization between liver cis-eQTL and GWAS signal was defined as showing a posterior probability of sharing the same SNP (PP4) of larger than 0.8. A lead SNP at the *SORT1/CELSR2* locus (rs629301) showed a positive colocalization signal, but the cis-eQTL co-localized with both genes, so SNP-based genetic effects for these genes could not be analysed separately.

**UK Biobank lipid and CAD phenotypes**. The UK Biobank is a prospective study of over 500,000 participants recruited at an age of 40–69 years from 2006 to 2010 in the United Kingdom. Participant data include health records, medication history and self-reported survey information, together with imputed genome-wide geno-types and biochemical measures[23]. Baseline biochemical measures including LDL cholesterol (LDLc), HDL cholesterol (HDLc), triglycerides (TG), and serum total cholesterol (TC) had been obtained in UK Biobank's purpose-built facility in Stockport as described in the UK Biobank online data showcase and protocol (www.ukbiobank.ac.uk). Demographic and other relevant phenotypic information was obtained from standard questionnaire data. Individual lipid phenotypes (LDLc, HDLc, TG and TC) were first modelled as dependent variables using linear regression models against covariates including age, sex, smoking, alcohol drinking status, BMI, lipid medication use and top ten genetic principle components. Residuals were then used as outcome variables in genetic analyses. Lipid medica-tion use was obtained from self-reported questionnaire data (UK Biobank fields 6153 and 6177). CAD cases were recognized based on both self-reported diagnosis and Hospital Episode Statistics data in the UK Biobank with a code-based CAD definition as presented in the most recent CAD GWAS that included UK Biobank[53]. In total, 30,125 CAD cases were identified and the CAD analysis was conducted using logistic regression models adjusted for age, sex, smoking status, alcohol drinking status, BMI, lipid medication use and top ten genetic principle components. All phenotype data were derived from the UK Biobank basket 'ukb27390' released on March 11, 2019.

**Pairwise gene-based PTV burden interaction testing**. High-impact protein-truncating variants (PTVs) expected to disrupt protein functions were identified from 302,331 whole-exome sequencing (WES) data of UK Biobank participants to conduct pairwise interaction analyses. WES data were generated and quality controlled (QCed) as described in Van Hout et al. at the Regeneron Genetics Center as part of a collaboration between AbbVie, Alnylam Pharmaceuticals, AstraZeneca, Biogen, Bristol-Myers Squibb, Pfizer, Regeneron and Takeda and the UK Biobank consortium[54]. PTVs were called from a Regeneron QC-passing 'Goldilocks' set of genetic variants using Variant Effect Predictor v96[26] and the LOFTEE plugin[25]. LOFTEE applies a range of filters on stop-gained, splice-site disrupting and frameshift variants in order to exclude putative PTVs due to variant annotation and sequencing mapping errors that are unlikely to significantly disrupt gene function. For instance, stop-gained and frameshift variants that are within 50 kb of the end of the transcript will be flagged as 'low-confidence'. For our analysis, we only considered variants predicted as PTVs and flagged as 'high confidence' by LOFTEE for each canonical transcript (as defined in Ensembl). We identified 573,369 high-confidence PTVs in the canonical transcripts of 19,076 genes. This set included 983 rare PTVs in the 30 lipid genes analysed in this study. PTVs per gene were enumerated, and a PTV burden association analysis was conducted in 240,970 unrelated (>2nd degree relatedness) UK Biobank participants of European ancestry, as defined by principle components analysis of the geno-typing data[23].

For pairwise PTV-based interaction testing, QCed UK Biobank lipid phenotypes (HDLc, LDLc, TG and TC) after residualization against the covariates were modelled as dependent variables using the following four robust linear regression models:

Model 1 for gene1 PTV burden only: lipids ~ $PTV_1$
Model 2 for gene2 PTV burden only: lipids ~ $PTV_2$
Model 3 for gene1 PTV burden and gene2 PTV burden (additive effects, AEs): lipids ~ $PTV_1 + PTV_2$
Model 4 for gene1 PTV burden and gene2 PTV burden (GI): lipids ~ $PTV_1 + PTV_2 + PTV_1 * PTV_2$

Schwarz's Bayesian Information Criterion (BIC)[58] scoring was used to determine the best model to explain the data and goodness of fit, with the lowest BIC value indicating the best-fitting model describing each gene pair. Model 3

reflected additive effects (AEs), Model 4 gene interactions (GIs). The model with the lowest BIC was chosen as describing most adequately the type of interaction between each corresponding gene pair. All models were fitted using the rlm() function in R package 'MASS'.

In order to also estimate statistical significance of gene interaction and additive effect for each PTV combination, we calculated normal distribution approximated $p$-values from the $t$-statistics of the robust linear regression full model (Model 3 or Model 4 depending on the gene pair), including the interaction term (if interaction testable), as well as the PTV effect for each single gene, for all gene pairs and all four lipid measures. To correct for multiple comparisons, the $p$-values were then adjusted using the false discovery rate (FDR) method[59], after which genetic interactions, as well as additive effects, were identified in a two-step manner: if the FDR-corrected $p$-value of the interaction term was less than 0.005 and the BIC indicated Model 4 as the best fitting model, then the gene pair was considered as being a significant GI; if the FDR-corrected $p$-value of the interaction term was larger than 0.01 or untestable, but FDR-corrected $p$-value for each single-gene PTV effect was less than 0.005 and the BIC indicates Model 3 as the best fitting model, the gene pair was considered as a significant AE.

**Pairwise SNP interaction testing**. To assess whether GWAS lead SNPs modulate plasma lipid levels through joint effects within and across GWAS loci, we con-ducted pairwise SNP-SNP interaction analysis using genome-wide genotyping data and biochemical measures of lipid species from the UK Biobank. Twenty-eight lead SNPs mapped to the 30 lipid GWAS genes were extracted from genotyping data of 378,033 unrelated (removed up to 2nd degree relatedness) participants of European ancestry. A total of 378 pairwise interaction effects were tested for each of the four lipids species after residualization against the covariates by running four robust linear models using rlm() function from R package 'MASS':
Model 1 for SNP1 only: lipids ~ $SNP_1$
Model 2 for SNP2 only: lipids ~ $SNP_2$
Model 3 for SNP1 and SNP2 (AE): lipids ~ $SNP_1 + SNP_2$
Model 4 for SNP1 and SNP2 (GI): lipids ~ $SNP_1 + SNP_2 + SNP_1 * SNP_2$

Schwarz's Bayesian Information Criterion (BIC) scoring was used to determine the best model to explain the data and goodness of fit, with the lowest BIC value indicating the best-fitting model describing each SNP pair. If Model 3 had the lowest BIC value, it reflected an AE, and if Model 4 had the lowest BIC value, it reflected a GI.

A similar strategy was applied for pairwise interaction testing to explore potential joint effects between the 30 genes on CAD risk by running the following four logistic regression models adjusted for age, sex, smoking status, alcohol drinking status, BMI, lipid medication use and ten principle components:
Model 1 for SNP1 only: CAD ~ $SNP_1$
Model 2 for SNP2 only: CAD ~ $SNP_2$
Model 3 for SNP1 and SNP2 (AE): CAD ~ $SNP_1 + SNP_2$
Model 4 for SNP1 and SNP2 (GI): CAD ~ $SNP_1 + SNP_2 + SNP_1 * SNP_2$

As above, the model with the lowest BIC was chosen as describing most adequately the type of interaction between each corresponding SNP pair.

We also calculated normal distribution approximated $p$-values from the $t$-statistics of the robust linear regression and logistic regression full model (Model 4), including the interaction term, as well as the SNP effect for each single gene, for all gene pairs and all four lipid measures as well as CAD. To correct for multiple comparisons, the $p$-values were then adjusted using the false discovery rate (FDR) method[59], after which genetic interactions, as well as additive effects, were identified in a two-step manner: if the FDR-corrected $p$-value of the interaction term was less than 0.005 and the BIC indicated Model 4 as the best fitting model, then the gene pair was considered as being a significant GI; if the FDR-corrected $p$-value of the interaction term was larger than 0.01 or untestable, but FDR-corrected $p$-value for each single-gene PTV effect was less than 0.005 and the BIC indicates Model 3 as the best fitting model, the gene pair was considered as a significant AE.

**PTV-SNP interaction testing**. In order to conduct pairwise interaction analyses between GWAS lead SNPs and PTVs, we assessed the interaction of the 28 lead SNPs with rare PTV burden for each of the 30 genes. For SNP-PTV interaction testing, UK Biobank lipid phenotypes (HDLc, LDLc, TG and TC) after residuali-zation against the covariates were modelled as dependent variables using the fol-lowing four robust linear regression models fitted by rlm() function in R package 'MASS':
Model 1 for gene1 lead SNP only: lipids ~ $SNP_1$
Model 2 for gene2 PTV burden only: lipids ~ $PTV_2$
Model 3 for gene1 lead SNP and gene2 PTV burden (AE): lipids ~ $SNP_1 + PTV_2$
Model 4 for gene1 lead SNP and gene2 PTV burden (GI):
lipids ~ $SNP_1 + PTV_2 + SNP_1 * PTV_2$

As above, the model with the lowest BIC was chosen as describing most adequately the type of interaction between each corresponding SNP-gene pair.

We also calculated normal distribution approximated $p$-values from the $t$-statistics of the robust linear regression full model (Model 3 or Model 4 depending on the gene pair), including the interaction term (if interaction testable), as well as the PTV or SNP effect for each single gene, for all gene pairs and all four lipids measurements. To correct for multiple comparisons, the $p$-values were then adjusted using the false discovery rate (FDR) method[59], after which genetic

interactions, as well as additive effects, were identified in a two-step manner: if the FDR-corrected p-value of the interaction term was less than 0.005 and the BIC indicated Model 4 as the best fitting model, then the gene pair was considered as being a significant GI; if the FDR-corrected p-value of the interaction term was larger than 0.01 or untestable, but FDR-corrected p-value for each single-gene PTV effect was less than 0.005 and the BIC indicates Model 3 as the best fitting model, the gene pair was considered as a significant AE.

**PTV–PRS interaction testing**. We assessed the interaction effects between poly-genic risk score (PRS) and PTVs for each of the four lipid phenotypes. To construct PRS for UK Biobank samples, we first derived the PRS weights for each SNP across the genome using PRS-CS[29], which is a Bayesian regression-based algorithm, and publicly available summary statistics from lipid GWAS[1]. We applied derived PRS weights to imputed genotypes (with minor allele frequency >0.01 and imputation quality INFO > 0.8) of UK Biobank samples and calculated PRS for each lipid, based on the corresponding PRS weights. Note that all SNPs in the gene of interest were excluded from the PRS when testing for PRS–PTV gene interaction. To test GIs between PRS and PTV burden for each of the 30 genes, UK Biobank lipid phenotypes (HDLc, LDLc, TG and TC) after residualization against the covariates were modelled as dependent variables by fitting the four robust linear regression models using rlm() function in R package 'MASS':
Model 1 for PRS only: lipids ~ PRS
Model 2 for gene PTV burden only: lipids ~ PTV
Model 3 for PRS and gene PTV burden (AE): lipids ~ PRS + PTV
Model 4 for PRS and gene PTV burden (GI): lipids ~ PRS + PTV + PRS * PTV

As above, the model with the lowest BIC was chosen as describing most adequately the type of interaction between each corresponding PRS-gene pair.

We also calculated normal distribution approximated p-values from the t-statistics of the robust linear regression full model (Model 4), including the interaction term, as well as the PRS and PTV effect, for all PRS–PTV pairs and all four lipids measurements. To correct for multiple comparisons, the p-values were then adjusted using the false discovery rate (FDR) method[59], after which genetic interactions, as well as additive effects, were identified in a two-step manner: if the FDR-corrected p-value of the interaction term was less than 0.005 and the BIC indicated Model 4 as the best fitting model, then the gene pair was considered as being a significant GI; if the FDR-corrected p-value of the interaction term was larger than 0.01 or untestable, but FDR-corrected p-value for each single-gene PTV effect was less than 0.005 and the BIC indicates Model 3 as the best fitting model, the gene pair was considered as a significant AE.

**Multiple testing correction with false discovery rate (FDR)**. We performed FDR-correction as described by Benjamini and Hochberg, 1995[59]. P-values were calculated based on t-values extracted from the rlm result for the fitted effects of single genes and interaction terms. For both genetics and primary coRNAi screen, we set the threshold to call a gene pair as GI to an FDR-corrected p-value for the interaction term as p < 0.005, with FDR correction being based on the number of interactions between gene/variant pairs tested (n = 1740 for PTV-PTV interaction, n = 1890 for SNP-SNP interaction, n = 3240 for PTV-SNP interaction, n = 120 for PTV–PRS interaction, and n = 435 for coRNAi primary screen and n = 36 for coRNAi validation experiments) times the number of genetic effects tested (n = 3, single-gene effects for gene1 and gene2, and the interaction effect if testable). FDR-correction was done separately for PTV-PTV, SNP-SNP, PTV-SNP, and PTV-PRS interaction for genetic analyses given the different nature of genetic variations tested, but a rather stringent significance cut off was chosen at FDR corrected p < 0.005 for all analyses. For the validation RNAi experiments, significance threshold was set at an FDR corrected p < 0.01.

**RNAi interaction testing**

*Cells and reagents.* HeLa-Kyoto cells are a strongly adherent Hela isolate (gift from S. Narumiya, Kyoto University Japan) that, as we demonstrated earlier, enable reliable measurements of LDL-cholesterol uptake dynamics and show lipid homoeostatic mechanisms similar to those described for liver-derived cell models[19,30,31]. DiI-LDL (Life Technologies), DRAQ5 (Biostatus), Dapi (Molecular Probes), 2-hydroxy-propyl-beta-cyclodextrin (HPCD) (Sigma), Lipofectamine 2000 (Invitrogen) and Benzonase (Novagen) were purchased from the respective suppliers.

*siRNA selection and production of siRNA microarrays.* RNA interference (RNAi) screening was conducted in glass-bottomed single-well chambered cell culture (Lab-Tek) slides with solid-phase reverse siRNA-transfection of cultured cells ('cell microarrays') as described previously[19,31,60]. Each gene under study was targeted with a single siRNA (Silencer Select, Invitrogen) that had been selected with the EMBL-generated software tool bluegecko (J.K. Hériche, in house database) based on the alignment to the reference genome, a maximal number of protein-coding transcripts per gene targeted and expected specificity for the target gene. The 28 siRNAs in this study had been validated earlier to significantly enhance or reduce cellular uptake of fluorescent-labelled LDLc (DiI-LDL) or free cellular cholesterol levels[19] and were shown to efficiently downregulate mRNA or protein levels of their respective target genes (Supplementary Data 2). siRNA sequences are

provided in Blattmann et al., 2013 Supplementary Data 4. For the two genes not analysed in our earlier study (*MYLIP*, *PAFAH1B2*), siRNAs used in the current study were prioritized from 3 and 5 siRNAs per gene based on bluegecko *in silico* analyses, knockdown efficiency on target mRNA/protein levels (up to less than 10% residual levels) and/or efficiency to modulate cellular DiI-LDL uptake in pre-paratory individual single gene knock-down experiments (not shown). The 75% (12/16) of siRNAs that had scored as individually modulating cellular DiI-LDL uptake in our earlier study[19] also met the more stringent criteria of our current study to score as LDLc uptake modulator when used either alone or together with non-silencing control siRNA Neg9 (Fig. 3b, CTRL column), thereby replicating our earlier results and validating experimental settings for this current study.

To cover the total of 435 pairwise siRNA combinations including controls and replicas, five different cell microarrays with 384 spots/array were produced. Per array, the following negative controls were added: eight spots containing *INCENP*-siRNA (s7424) to control for transfection efficiency[19]; eight spots containing non-silencing control siRNA Neg1 (s229174), and eight spots containing non-silencing control siRNA (denoted as CTRL throughout the text) Neg9 (s444246). Furthermore, eight spots were added with siRNA targeting *LDLR* (s224006) as a positive control for LDLc uptake, as well as eight spots with siRNA targeting *NPC1* (s237198) knockdown of which increases free cellular cholesterol signals[30]. For pairwise combinatorial RNAi-screening, siRNAs against two genes were printed simultaneously on a respective siRNA-spot, with equal amounts (~0.053 pmol/siRNA) of siRNA per gene. As positive controls, eight spots containing both, non-silencing control siRNA Neg9 (CTRL) (s444246) and siRNA targeting *LDLR* (s224006), and eight spots containing both, non-silencing control siRNA (CTRL) Neg9 (s444246) and siRNA targeting *NPC1* (s237197) were included per array. For all genes, 'single-gene knockdown' scenarios [siRNA$_{geneA}$ + Neg9] were added on two spots per array. Each pairwise 'combinatorial knockdown' scenario [siRNA$_{geneA}$ + siRNA$_{geneB}$] was analysed on one spot per array, with a single spot covering 50-100 informative cells[31,61] (Supplementary Fig. 2).

In order to confirm genetics interactions identified with the coRNAi screen, we replicated our analyses with forward transfection in a liquid-phase format with Lipofectamine 2000 reagent in 12-well plates, according to the manufacturer's instructions. Concentrations of the siRNAs were adjusted to mimic the single knockdown phenotypes from the screen (Supplementary Data 2). 1 μl of Lipofectamine 2000 was used per each transfection. GIs that showed statistically significant interaction effects ($p_{FDR} < 0.01$) in replication analyses and for which interaction values showed the same as in the primary coRNAi screen, were considered as validated (Supplementary Data 11).

**Cell culture, transfection and LDLc uptake assay**. HeLa Kyoto cells were grown in DMEM medium (Gibco) supplemented with 10% (w/v) foetal calf serum (FCS) (PAA) and 2 mM L-glutamine (Sigma) at 37 °C with 5% CO$_2$ and saturated humidity. Cells were plated at a density of $6 \times 10^4$ per plate on the cell microarrays for solid-phase siRNA transfection[60] and cultivated for 48 h before performing the LDLc uptake assay. For liquid phase transfection-based validation experiments, cells were plated in 12-well plates the day prior to transfection, and siRNA-transfected cells were cultivated for 48 hours. The assays to monitor cellular uptake of fluorescently-labelled LDLc (DiI-LDL) were performed as described also in the previous publications[19,30,31]. Cells cultured in serum-free medium (DMEM/2 mM L-glutamine/0.2% (w/v) BSA) and exposed to 1% 2-hydroxy-propyl-beta-cyclo-dextrin for 45 min were labelled with 50 μg/ml DiI-LDL (Invitrogen) for 30 min at 4 °C. DiI-LDL uptake was stimulated for 20 min at 37.0 °C. Endocytosis of labelled LDLc was stopped by washing with ice-cold media. We removed non-internalized DiI-LDL from the plasma membrane by acidic wash for 1 min in medium at pH 3.5 performed also at 4 °C. This was followed by fixation, counterstaining for nuclei (Dapi) and cell outlines (DRAQ5). For RNAi-based gene interaction screening, each of the five cell microarrays was assayed in 7–10 biological replicas.

**Image acquisition and quality control**. Image acquisition was performed using an Olympus IX81 automated microscope with Scan^R software and an UPlanSApo 20×/0.4NA air objective as described[19,31]. Images from a total of 42 cell micro-arrays were visually quality controlled. Arrays with insufficient knockdown efficiency where *INCENP* siRNA treated cells did not show the expected multinucleated phenotype in the DAPI channel were excluded. Also, arrays with plate effects as evaluated through diagnostic plots with the *splot* function in R, and arrays where knockdown of *LDLR*, or *LDLR* together with negative control siRNA Neg9, did not show a significant difference from controls, were discarded as well. Following these QC criteria, 29 cell microarrays with a total of 11,047 image frames per channel were further analysed. The in-house developed tool HTM Explorer (C. Tischer; https://github.com/embl-cba/shinyHTM) was then used to select images fulfilling pre-defined criteria for cell number, image sharpness quality, and image background intensity, resulting in a total number of 9539 (86.35%) QCed image frames that were used for subsequent analyses.

**Image analysis**. Automated image analysis was performed using a specifically developed pipeline (available upon request) in the open-source software CellProfiler[62] http://www.cellprofiler.org as described[19,31]. In brief, areas of indi-vidual cells were approximated by stepwise dilation of masks on the DAPI (nuclei)

and DRAQ5 (cell outlines) channels[63]. For each individual cell, DiI-LDL signal was determined from masks representing intracellular endosome-like vesicular areas that were determined by local adaptive thresholding according to predefined criteria for size and shape (Supplementary Fig. 2). Initially, the total fluorescence intensity of DiI-signal above the local background per cell mask was quantified, and mean values were calculated from all cells per image. Then, for each single siRNA ([siRNA$_{geneA}$ + Neg9] and [siRNA$_{geneB}$ + Neg9]), or double siRNA knockdowns ([siRNA$_{geneA}$ + siRNA$_{geneB}$]), these mean values from different images from the same biological replicate were averaged ([$I_F treated$] and a robust Z-score was calculated in HTM Explorer using the median of total fluorescence signal of all the negative control siRNAs per array ([$medianI_F(controls)$]) and the median absolute deviation of these controls ($madI_F(controls)$) as follows[64,65]:

$$robustZ - score = \frac{I_F treated - medianI_F(controls)}{madI_F(controls)} \quad (1)$$

A median robust Z-score was calculated per treatment across all biological replicates and is represented on the heatmap (Fig. 3c) and in the respective Supplemental tables (Supplementary Data 10–11).

**RNAi gene interaction testing**. To identify pairs of genes for which simultaneous knock-down results in additive effects (AEs) or gene interactions (GIs) on LDLc uptake we conducted a Robust Linear Model fitting in R (rlm() function of 'MASS' package). Robust Z-score values calculated from different biological replicates in the presence of single ([siRNA$_{geneA}$ + Neg9] and [siRNA$_{geneB}$ + Neg9]) or double knock-downs ([siRNA$_{geneA}$ + siRNA$_{geneB}$]) were considered to be response variable values. Negative control values [Neg9] were included in each fitted dataset to correctly account for baseline LDLc uptake. The full regression model considered in the study was

$$y = \beta_0 + \beta_A * x_A + \beta_B * x_B + \beta_{AB} * x_A * x_B + \epsilon \quad (2)$$

which is equivalent to the short form of the statistical formula:

$$y \sim x_A + x_B + x_A * x_B \quad (3)$$

In both formulas $y$ corresponds to the robust Z-score values of measured LDLc uptake; $x_A$, $x_B$ are encoded predictor variables, which are equal to 1 in case of presence of siRNA$_{geneA}$, siRNA$_{geneB}$, or both siRNAs accordingly and equal 0 otherwise. The $\epsilon$ is a noise term, which is minimised during the fitting process. Model fitting provides estimates of $\beta_0$, $\beta_A$, $\beta_B$ and $\beta_{AB}$ values. $\beta_0$ defines the effect of the negative control on robust Z-score values and can be also denoted as an intercept of the linear fit. For our data $\beta_0$ is always close to 0 because of the robust Z-score definition. The $\beta_A$ and $\beta_B$ define individual effects of siRNA$_{geneA}$ and siRNA$_{geneB}$ accordingly. The $\beta_{AB}$ defines the interaction effect, denoted in the text, figures and tables as Interaction Value, between genes A and B and represents the difference between the observed robust Z-score values in case of double knockdown $y_{AB}$ and the expected additive effect of geneA and geneB knockdown ($\beta_{AB} = y_{AB} - \beta_0 - \beta_A - \beta_B$).

Subsequently, two strategies were used to evaluate functional interactions for each gene pair using a defined statistical model:

First, we used primary screen data to identify likely gene pairs for which GIs and AEs observed upon combinatorial knockdowns. For this we compared fitting of the whole model to the fitting of reduced model versions. Following models were compared:

Model 0 - (only baseline effect $\beta_0$ in case of either single or double knockdown): $y \sim 1$

Model 1 - effect of siRNA$_{geneA}$ only: $y \sim x_A$

Model 2 - effect of siRNA$_{geneB}$ only: $y \sim x_B$

Model 3 for additive effect of both siRNAs (AE): $y \sim x_A + x_B$

Model 4 – full model including genetic interaction (GI): $y \sim x_A + x_B + x_A * x_B$.

To determine the best model explaining the data for each gene pair we used Schwarz's Bayesian Information Criterion (BIC)[58]. BIC score was calculated for each model fitted to the data, then the model with the lowest BIC value (BIC*) was selected as the best-fitting model. Co-knockdown effects of each gene pair were classified as AEs or GIs when model 3 or model 4 accordingly were defined to fit data best. Additionally, for the coRNAi screen, we used the method published by Raftery, 1995 to define the strength of evidence for the respective model to be selected[66]. Namely, if the difference (ΔBIC) between the BIC value of the best fitting model (the model with the lowest BIC value) and the BIC value of any other model is bigger than 2, then it would indicate a significant evidence for this model (with BIC*) to truly represent the data. In other words, if ΔBIC > 2 then the model with lowest BIC value (BIC*) was considered as the one most correctly describing the data in comparison to other tested models. If the ΔBIC < 2, then two models were considered as possible alternatives for representing the dataset.

Secondly, to estimate the statistical significance of gene interaction effect for each siRNA gene combination and to confirm potential AE and GI pairs identified by the BIC method, we calculated a p-value from the t-value of the linear regression model terms describing effects of individual gene knockdowns ($\beta_A$ and $\beta_B$) and genetic interaction ($\beta_{AB}$) as $p_{Val} = 2 - 2 * p_{norm}(abs(t_{Val}))$. To correct for multiple comparisons, the p-values were adjusted using the false discovery rate (FDR) method[62]. We applied FDR-correction on the pulled together p-values of individual gene knockdown effects ($\beta_A$ and $\beta_B$) and genetic interaction term ($\beta_{AB}$),

thus we calculated FDR-correction on the set of $3 \times 435 = 1305$ p-values. The FDR-corrected p-values $p_{FDR} < 0.005$ were considered to correspond to significant GIs.

AEs were defined as pairs for which both individual effect terms ($\beta_A$ and $\beta_B$) were significant having a $p_{(FDR)} < 0.005$, whereas the interaction term ($\beta_{AB}$) was not significant, namely $p_{(FDR)} > 0.01$. Both were defined only when BIC analysis indicated Model 3 and each of the SNPs separately had significant effect.

36 gene pairs were subsequently taken for validation experiments by liquid phase transfection. Of these, we considered 20 to be validated when $p_{(FDR)} < 0.01$ and Interaction Value had the same directionality ('−' or '+') as in the primary screen.

GIs identified through coRNAi were classified (according to the review article Boucher B., Jenna S., 2013 and EMBL-EBI ontology website: https://www.ebi.ac.uk/ols/ontologies) as negative—aggravating or synergistic, i.e., simultaneous knockdown of both genes magnified the effect size beyond expectations for an additive effect and positive GIs. Positive GIs were further subdivided into: alleviating, i.e., the joint effect was approximately equal to the most severe of the phenotypes and suppressive, i.e., the joint effect was less severe ('healthier' or 'closer to wild-type') than the most severe of the phenotypes[22].

**RT-qPCR analysis**. Cell lysis and total RNA extraction was done using the RNease Mini Kit (Qiagen). Reverse-transcription was performed with the SuperScript™ III First-Strand Synthesis SuperMix for RT-qPCR (Invitrogen). Raw data was collected with StepOne Software v2.3. RT-qPCR data were obtained from three biological replicates/siRNA treatment. For each siRNA treatment target mRNA was normalized to that of *GAPDH* and compared to CTRL siRNA and the log2 of fold change ($2^{-\Delta\Delta CT}$) was calculated (see Supplementary Fig. 6). Results were considered as significant if p-values were below 0.05 in a two-tailed Student's t-test. Primer sequences are provided in Supplementary Table 1.

**Immunocytochemistry and confocal microscopy**. Cells were transfected and cultured as described above then fixed with 3% paraformaldehyde (PFA) at room temperature for 20 min, washed with PBS, incubated with 30 mM glycine for 5 min and washed again with PBS. For LDLR staining cells were permeabilized with 0.05% Filipin III (Sigma #F4767) in 10% FCS for 30 min at room temperature. Primary antibody: rabbit monoclonal anti-LDLR (Fitzgerald #20R-LR002) was diluted 1:100 in 5% FCS overnight at 4 °C. Secondary antibody: goat polyclonal goat anti-rabbit IgG Alexa 568 (Invitrogen #A11011) was diluted 1:400 in 5% FCS. Fixed cells were imaged using a Zeiss LSM 780 confocal microscope using a 63×/1.4NA oil immersion objective. For plasma membrane LDLR staining, cells were not permeabilized, but directly stained for 1 hour at 4 °C with antibody recognising the extracellular part of LDLR, namely, mouse anti-LDLR-C7 antibody (Progen #61087) diluted 1:100 in PBS. As a secondary antibody, we used chicken anti-mouse IgG Alexa 488 (Invitrogen #A-21200) diluted 1:400. Cells were imaged on an Olympus FluoView 3000 confocal microscope using 60×1.3NA silicon oil immersion objective.

**Reporting summary**. Further information on research design is available in the Nature Research Reporting Summary linked to this article.

## Data availability

Genetic data utilized in this study have been or will be made publicly available via UK Biobank (for updates, see here: https://www.ukbiobank.ac.uk/enable-your-research/about-our-data/genetic-data). A 'minimum dataset' of original image files, the Cell Profiler pipeline and all R-codes required to fully recapitulate how results were generated from coRNAi-analyses in a step-by-step manner have been deposited at https://git.embl.de/grp-almf/genetic-interactions-screen-lipid-levels. All source imaging data are deposited at https://idr.openmicroscopy.org/about/. Additional data that support the findings of this study can be made available through contacting the corresponding author Heiko Runz (heiko.runz@gmail.com). Source data are provided with this paper.

## Code availability

Computational codes for all interaction analyses (robust linear model fitting, BIC model selection and FDR correction) in the UK Biobank and on RNAi experimental data are available at: https://git.embl.de/grp-almf/genetic-interactions-screen-lipid-levels.

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

## Acknowledgements

This research has been conducted using the UK Biobank resource under application number 26041. We thank all the participants and researchers of UK Biobank for making these data open and accessible to the research community. AbbVie, Alnylam

Pharmaceuticals, AstraZeneca, Biogen, Bristol-Myers Squibb, Pfizer, Regeneron and Takeda are acknowledged for generation and initial quality control of the whole-exome sequencing data. We thank Eric Marshall, Yongsheng Huang and Frank Nothaft for infrastructure support for genetic data analyses. The EMBL Advanced Light Microscopy Facility is acknowledged for supporting high-content microscopic-based screening analyses. We are grateful to Brigitte Joggerst, Susanne Theiss and Miriam Reiss for excellent technical assistance. Support for the study came in part from the Transatlantic Networks of Excellence Program 10CVD03 from Fondation Leducq to H.R. and R.P. M.Z. was supported by the EMBL EIPOD programme, A.T. and P.B. by the EMBL PhD programme.

## Author contributions

Conceptualization, H.R. and R.P.; Methodology and investigation, M.Z., Y.H., A.T., A.H., C.-YC., R.P. and H.R; Formal analysis and validation, M.Z., Y.H., A.T., C.-Y.C., J.L., A.H., B.K., E.T. and H.R; Resources and data curation, J.L., P.B., C.W., D.S., S.J., E.T., R.P. and H.R.; Writing—original draft, M.Z., A.T. and H.R; Writing—review and editing, M.Z., Y.H., A.H., P.B., E.T., R.P. and H.R.; Supervision, S.J., W.H., E.T., R.P. and H.R.; Project administration and funding acquisition, S.J., R.P. and H.R.

## Competing interests

Y.H., C.-Y.C., J.L., C.W., D.S., S.J. and H.R. are full-time employees at Biogen, Inc. The funders had no role in study design, data collection and analysis, decision to publish, or preparation of the manuscript. Other authors declare no competing interests.
