## [Peer Review File · Nature Communications]

Pairwise effects between lipid GWAS genes modulate lipid plasma levels and cellular uptakeREVIEWER COMMENTS

Reviewer #1 (Remarks to the Author):

General comments

The manuscript "Pairwise genetic interactions modulate lipid plasma levels and cellular uptake" by Zimon et al takes on a topic of both fundamental and clinical interest: how do genetic perturbations combine to affect lipid transport in humans.

On a fundamental level, it is of great interest to know whether genetic interaction, a phenomenon defined by two genetic changes that produce a phenotype that departs from expectation under a model of independent action, are as prevalent in vivo in humans as it has been found to be in human and yeast cell-based models. On a clinical level, the authors make the important point that combinatorial genetic effects could prove useful in selecting drug combinations. The project is especially timely, in that it uses UK Biobank exome data that has only recently been of sufficient size to contemplate this analysis.

Focusing on a set of genes identified by GWAS as being at loci where variation correlates with blood lipid levels, the authors leverage three parallel types of genetic information to uncover genetic interactions: protein-truncating variants (PTVs), common variants, and RNAi perturbation. There is a strong rationale for the study, and the observations about genetic interaction, if accurate, would be quite exciting.

However, I have several major concerns about the methodologies used, leading to the broader concerns that insufficient evidence is provided to reach specific confident conclusions about any individual genetic interaction.

Major issues

1) Although 'just' a language issue, the authors should clarify their language around genetic interaction and avoid describing non-interacting gene pairs as "additive genetic interactions". The field of genetic interactions is already full of confusingly divergent definitions for the same term (e.g., PMID: 9649511), leading many to abandon the term epistasis in favor of "genetic interaction" (defined by a combined effect that departs from independent expectation), with subtypes "negative genetic interaction" (more extreme than expected from independent action) and "positive genetic interaction" (less extreme than expected given independent action). Text would be more informative if all uses of nGI were replaced with the more specific "positive GI" or "negative GI".

1) Positive genetic interactions are often further divided into two or more categories (I will describe the most important two of these, but the authors can choose from a bewildering menagerie of alternative types and terms in the literature for these categories): "diminishing returns" positive interactions, where the double-perturbation phenotype is as extreme or less extreme than the most extreme single-perturbation phenotype, and "suppressive" positive genetic interaction, where the double-perturbation phenotype is less extreme than the most extreme single-perturbation phenotype. It would be useful if, where possible, cases of positive interaction could be separated into "diminishing returns" or "suppression".

2) Although absence of a genetic interaction (what the authors call "additive genetic interaction") would normally be considered the default scenario which is not worthy of much attention, the authors make an excellent case that non-interaction is interesting (whether the expected result or not, it suggests that a combination of drugs could have a greater effect than either alone). However, this highlights the point that, instead of just looking for negative interactions or non-interacting gene pairs, they should explicitly be identifying pairs that together yield a phenotype that is less extreme than the least extreme single mutant (or more extreme than the most extreme single mutant, depending on which direction away from wild-type is clinically interesting).

3) The authors use a model selection approach (with a fairly standard BIC implementation), which is essentially asking "if I had to guess what the genetic relationship is, what would my best guess be".

This approach might be fine for picking a working hypothesis, but it is a completely inadequate standard of evidence for claiming with any confidence that an interaction (or non-interaction) really exists. The authors should find some way of estimating confidence in their conclusions, e.g. showing that confidence intervals for the double-perturbation effect (e.g., obtained from PTV data by bootstrap resampling from subjects) shows a departure in the right direction from the relevant single gene effect.

4) The authors only provide measures of statistical significance for the RNAi approach. This calculation is flawed, in that the authors seem to be (line 852) assuming that a t-statistic follows a normal distribution. Given that t distributions typically have fatter tails than normal distributions, this approach will provide misleadingly low measures of significance. Perhaps their FDR procedure accounts for this, but no details are given on the procedure (e.g., what was the source of null distribution for this calculation).

5) Interpretation of LDL uptake data should consider that fluorescence measures of DiI-LDL will include LDL bound to the cell surface (e.g., line 224-5 the higher DiI-LDL signal in LDLRAP1 knockdown may also be interpreted as more binding to LDLR on cell surface (makes sense since LDLRAP1 is required for positioning LDLR at clathrin-coated pits). The suggested positive GI between LDLR and LDLRAP1 seems unfounded. LDLRAP1 functions in the internalization of the LDLR-LDL complex, and it has been well established that absence of LDLRAP1 results in accumulation of LDLR at the cell surface. The authors use a fluorescent LDL molecule (DiI-LDL), which does not distinguish between cell-surface bound and internalized LDL molecules, therefore while this may be a buffering interaction in that loss of LDLR and LDLRAP1 leads to a similar phenotype as loss of LDLR, the results of the co-RNAi screen are perhaps being misinterpreted.

6) Lines 188-191: The p.S447Ter stop-gain mutation in LPL is the only stop-gain variant in LPL known to have a gain-of-function effect (see PMIDs 1752947; 16574898) . As such, it would likely not be expected for other deleterious stop-gains in LPL to mitigate a high polygenic risk for elevated TG. On the contrary, loss of LPL function would be expected to have the opposite effect.

7) There should be discussion (both in general and specifically for each relevant finding) of the issue that GWAS SNPs cannot generally be unambiguously mapped to a single gene, so that the association may stem from a causal variant from another gene that is in LD with variants in the specific genes discussed.

Minor issues

2) Re: exome data analysis, it was not made clear if truncations appearing late in the coding region (which are often not damaging) were excluded. In addition to PTVs, the authors might have also considered missense variants that were confidently predicted as damaging by a computational predictor (although this would require some calibration against a sufficient number of known pathogenic and benign variants not used in training the predictor, which may not be available for all 30 genes of interest).

3) Lines 141-147: Should clarify what is the reduction in cholesterol due to the additive genetic interaction between PCSK9 and APOB. What baseline is the reduction of 40.01 mg/dl from, the average level of single gene carriers, this is not clear.

4) Lines 165: The authors mention the SNP rs4420638, which is located in APOC1, a gene highly implicated in CHD. The SNP is in linkage disequilibrium with SNPs in APOE. If there are suggested cis-effects between this SNP and the locus in question, the biology of this interaction should be discussed, or at minimum the causal gene mentioned, as this SNP is associated with increased TC, LDL and TG levels.

5) Lines 342-344: It is unclear how you would expect to detect the effects of a protein functioning in the trans-endothelial migration of leukocytes in this co-RNAi screen examining LDL-uptake (or surface-binding).

6) Lines 48-50: "For the vast majority of dyslipidemic individuals, however, no single-gene mutation can be identified." Should clarify that monogenic single variants could have been missed for many reasons, e.g., coding VUS or non-obvious splice-altering or other non-coding variants.

7) Figure 1. b & c: Statistical tests needed. Should also include plot for interaction between LPL-APOB on TG levels.

- 8) Lines 42-43: As worded, implies that only heritability influences blood lipids and CAD and neglects known environmental influences
- 9) Lines 68-69: Should cite substantial earlier literature that GIs can assist in nominating drug pair therapeutics.
- 10) Box and whisker plots don't do a good job of conveying subtle differences in the position or shape of distributions. Authors should consider representing/comparing distributions as histograms.

Reviewer #2 (Remarks to the Author):

Overall:

Zimon M and Runz H and colleagues evaluate genetic interactions between candidate lipid-associated alleles and compare to a combined RNA interference assay. They are motivated to discover drug targets for lipids that may be particularly effective in combination. Human genetic analyses occur in the UK Biobank with privileged access to 240K whole exome sequences as well as available GWAS data. They observe that much of the effects are independent/additive with few interactions. Two interactions appear to be consistent in RNA interference experiments. I have several concerns regarding the methods, and particularly the claims in the present study.

Major:

1. The design of this study is motivated by identifying gene interactions for known lipid genes to discover optimal drug combinations. Each of these genes are previously known to contribute to lipid variation independently, and this additively, already. The authors cite that statins as well as other cholesterol-lowering medicines lower LDL-C levels and CAD risk beyond statin treatment alone as a motivation – that's simply because these are independent targets. This is not a property of a uniquely synergistic effect. Therefore, the "additive" gene interactions identified are of unclear significance. I also worry that using the phrasing "gene interactions" actually implies a significant interaction (i.e., non-additive effect) and may be misleading. It may be best only describe non-additive interactions in this paper and remove additive effects- this latter aspect merely leads to confusion and is not novel.
2. A major limitation of the present work is the restriction just to 30 known lipid genes that are known to independently contribute to lipid variation. Therefore, the likelihood for identifying non-additive interactions is low. Extending beyond these genes and potentially using known protein-protein interaction data is more likely to yield novel insights.
3. The combinatorial RNA interference readout only measures LDL uptake into cultured cells. This is only one aspect of LDL cholesterol regulation. Others not assayed include apoB-containing lipoprotein synthesis in the liver, VLDL to LDL conversion via LPL and HL, transfer of cholesterol between HDL and LDL via CETP, etc. This substantially hinders the ability to identify new insights even in the candidate list of 30 known independent lipid genes.
4. Could non-additive interactions be driven by cryptic relatedness? Incorporating random effects of the genotypes would be helpful. Or, more simply, assessing correlation for the variants assessed would be helpful to address this concern.
5. Could non-additive interactions be driven by selection? If two independent alleles separately contribute to a deleterious phenotype (such as increased LDL-C), if they happen to co-occur in the same individual who is a healthy UK Biobank phenotype, this could lead to an apparently less severe phenotype in this individual than expected.
6. I'm not clear how common variant SNPs were selected. The methods indicate: "Twenty-eight lead SNPs mapped to the 30 lipid GWAS genes were extracted..." How were these connected biologically with each of the 30 lipid GWAS genes? Many of these loci also have locus heterogeneity but it appears this was not considered.
7. 4 models for several pairwise interaction tests for 4 traits each pairing are described throughout the

methods but criteria for statistical significance accounting for multiple hypothesis testing are not described in the methods. This comment applies to both the human genetic analyses as well as the RNAi analysis. As such, I'm not quite clear which of the present analyses are truly statistically significant if any based on the interaction p-values.

8. In the PTV gene-burden interaction analyses, why are LPL-APOB or PCSK9-APOB considered significant interactions? P-values for non-additive interactions for 3 of the 4 tests are >0.05 , and the 4th appears to not satisfy Bonferroni correction. I believe this claim is for additive effects. Showing independent effects for these is not novel and is confusing when using the term "interaction" when the p-values for interaction are nonsignificant. It's not surprising that APOB PTVs + PCSK9 PTVs lead to reduced cholesterol in an additive fashion and is likely not uniquely informative for a drug development program. Please refer to major point #1.

9. A number of the SNP-SNP interactions identified are in very close genomic proximity (e.g. rs2228603-rs58542926, rs118147862- rs4420638, rs118147862- rs2075650, etc). Is this a byproduct of LD and not reflective of true biological interaction? This is particularly true at the APOE locus where the authors claim there are several non-additive genetic interactions.

10. Across all the human genetic interaction testing and coRNAi interaction testing, only two non-additive interactions were consistent in both sets of experiments (SORT1-TOMM40 and NCAN-TOMM40). Identification of these may be novel. But given all the various testing, I'm not sure that these survive Bonferroni correction. If they do not, additional replication data would be helpful.

11. SORT1-TOMM40 and NCAN-TOMM40 are the prioritized interactions as noted above but significance is unclear. These are "buffering" interactions, meaning it would be inefficient to therapeutically target both together?

12. A paragraph of study limitations should be included in the Discussion.

Minor:

1. Page 1, line 31: "...not always translate to humans" should be "...do not always translate to humans."

2. Several words are combined with hyphens. This appears unnecessary in several instances. e.g., "CAD-risk," "statin-treatment," "exome-sequencing," "RNA-interference," "LDL-uptake," "QC-ed."

3. "LDL" is used for low-density lipoprotein cholesterol and "HDL" for high-density lipoprotein cholesterol. Better terms would be "LDL-C," "LDLc," "LDL cholesterol," etc or "HDL-C," "HDLc," "HDL cholesterol," etc, respectively.

4. The first introduction paragraph regarding monogenic and polygenic basis of lipids among dyslipidemic individuals currently reads as unrelated to the scientific premise of the present study. I would consider starting with the motivation of the study up front for better orientation.

5. In the methods, on page 23, there is a sentence "In total, 30,125 CAD cases identified and the cohort was adjusted for age, sex, smoking status, alcohol drinking status, BMI and lipid medication use." What was adjusted? What is the outcome and what's the exposure? Were principal components considered?

6. "Synergistic" and "buffering" should be defined in the Methods.

7. Page 5 line 119-120. The authors indicate "we sequenced the exomes of 200,654 UK Biobank participants..." If the authors are not going to be describing the exome sequencing methods, then better phrasing might be "we obtained the exome sequences of 200,654 UK Biobank participants..."

8. On page 5 line 123 – page 6 line 127. Most PTVs in LDLR, PCSK9, APOB have a phenotype consistent with familial hypercholesterolemia. This is likely the case for LDLR. But most PCSK9 or APOB PTVs have reduced lipids and that's why most of these are not listed as pathogenic in Clinvar.

Reviewer #3 (Remarks to the Author):

The authors looked for genetic interactions in blood lipid and coronary artery disease (CAD) traits in a large cohort, by mainly focusing on 30 significant GWAS hits that were previously functionally validated by the same group using RNAi knock down. Using exon sequencing, RNAi knock-down and available genome sequencing data, they sought for rare-common variants interactions (PTV-SNP),

common-common variants interactions (SNP-SNP), and interactions between pairwise RNAi knock downs. In summary, they found that these traits related to lipid metabolism is mainly driven by additive effect, and non-additive genetic interactions are rare and only contribute marginally to the traits.

Comments:

- The definition of "additive genetic interaction (aGI)" is misleading. What the authors testing here is the additive genetic affect, not a GI, as they defined themselves in figure 1. The authors should reword this in their manuscript.

- Related to the first point, all results and analyses regarding "aGI" are just additive genetic effects. The only results that are relevant to genetic interactions are the "naGI" data as defined by the authors.

- When talking about genetic interactions (naGIs), I don't have a clear sense of the magnitude of beta related to interactions, vs beta related to additive effects. I think it would be helpful for the authors to present this information in a more straightforward manner.

- The RNAi experiments are important here to address pairwise genetic interactions among the 30 validated causal genes. And indeed, some significant genetic interactions were found. However, it's not clear to me how the scores were determined and why do they sometimes switch signs? (Figure 3c)

This manuscript does not really expand our knowledge concerning the genetic architecture of complex traits. However, it is likely relevant to researchers interested in lipid metabolism and CAD.

Zimon et al., Response to the Reviewers' comments:

We would like to thank the Reviewers for their thoughtful comments and suggestions, which we think have significantly improved our manuscript. Please find below our point-by-point answers and an outline of revisions introduced in the text. We also would like to highlight that all scripts established and used for the analyses described in the manuscript are now available at: <https://git.embl.de/grp-almf/genetic-interactions-screen-lipid-levels>

Reviewer #1 (Remarks to the Author):

General comments

The manuscript “Pairwise genetic interactions modulate lipid plasma levels and cellular uptake” by Zimon et al takes on a topic of both fundamental and clinical interest: how do genetic perturbations combine to affect lipid transport in humans.

On a fundamental level, it is of great interest to know whether genetic interaction, a phenomenon defined by two genetic changes that produce a phenotype that departs from expectation under a model of independent action, are as prevalent *in vivo* in humans as it has been found to be in human and yeast cell-based models. On a clinical level, the authors make the important point that combinatorial genetic effects could prove useful in selecting drug combinations. The project is especially timely, in that it uses UK Biobank exome data that has only recently been of sufficient size to contemplate this analysis.

Focusing on a set of genes identified by GWAS as being at loci where variation correlates with blood lipid levels, the authors leverage three parallel types of genetic information to uncover genetic interactions: protein-truncating variants (PTVs), common variants, and RNAi perturbation. There is a strong rationale for the study, and the observations about genetic interaction, if accurate, would be quite exciting.

However, I have several major concerns about the methodologies used, leading to the broader concerns that insufficient evidence is provided to reach specific confident conclusions about any individual genetic interaction.

Major issues

1) Although ‘just’ a language issue, the authors should clarify their language around genetic interaction and avoid describing non-interacting gene pairs as “additive genetic interactions”. The field of genetic interactions is already full of confusingly divergent definitions for the same term (e.g., PMID: 9649511), leading many to abandon the term epistasis in favor of “genetic interaction” (defined by a combined effect that departs from independent expectation), with subtypes “negative genetic interaction” (more extreme than expected from independent action) and “positive genetic interaction” (less extreme than expected given independent action). Text would be more informative if all uses of nGI were replaced with the more specific “positive GI” or “negative GI”.

Positive genetic interactions are often further divided into two or more categories (I will describe the most important two of these, but the authors can choose from a bewildering menagerie of alternative types and terms in the literature for these categories): “diminishing returns” positive interactions, where the double-perturbation phenotype is as extreme or less extreme than the most extreme single-perturbation phenotype, and “suppressive” positive genetic interaction, where the double-perturbation phenotype is less extreme than the most extreme single-perturbation phenotype. It would be useful if, where possible, cases of positive interaction could be separated into “diminishing returns” or “suppression”.

We thank the Reviewer for this very helpful guidance on terminology which we agree is highly diverse in the genetic interaction field and at times inconsistent across genetics, *in vivo* and *in vitro* biology. In our revised manuscript we have followed the Reviewer’s recommendations and now more succinctly distinguish the different types of genetic effects assessed in our study. For instance, pairwise effects

that in our original manuscript had been termed “*additive genetic interactions*” or “*aGIs*” have now been renamed to “*additive effects*” and “*AEs*”. “*Non-additive GIs*” are now just “*GIs*” which we have further classified as either positive (alleviating or suppressive) or negative (aggravating) GIs. This terminology is based on an article by Benjamin Boucher and Sarah Jenna (2013; PMID: 24381582) and follows recommended terminology of EMBL-EBI ontology search:

https://www.ebi.ac.uk/ols/ontologies/mi/terms?iri=http%3A%2F%2Fpurl.obolibrary.org%2Fobo%2FMI_0208&viewMode=All&siblings=false).

We introduce the new terminology in the Results section, page 4 as such: “*For each gene pair, both the additive effects (AEs) (model 3), defined by the sum of effects from each gene or variant individually, as well as the genetic interactions (GIs) (model 4), represented by observed effects different than the expected additive effect, were calculated, with GIs being divided further into negative (aggravating) or positive (either alleviating or suppressive) (Figure 1a and Methods)*”. We have further modified Figure 1A according to this terminology and also adjusted the title of our manuscript which now reads: “*Pairwise effects between lipid GWAS genes modulate lipid plasma levels and cellular uptake*”. The revised terminology is being used throughout the text, figures and tables.

2) Although absence of a genetic interaction (what the authors call “additive genetic interaction”) would normally be considered the default scenario which is not worthy of much attention, the authors make an excellent case that non-interaction is interesting (whether the expected result or not, it suggests that a combination of drugs could have a greater effect than either alone). However, this highlights the point that, instead of just looking for negative interactions or non-interacting gene pairs, they should explicitly be identifying pairs that together yield a phenotype that is less extreme than the least extreme single mutant (or more extreme than the most extreme single mutant, depending on which direction away from wild-type is clinically interesting).

We fully agree with the Reviewer that not only (non-additive) genetic interactions (GIs), but also additive effects (AEs) can be considered as highly informative, for instance when prioritizing gene pairs as putative drug targets for combination therapies, which is one of the reasons why we have studied AEs in this manuscript. From our perspective an excellent example for this is the AE identified through our exome-based PTV-burden analysis and presented in Figure 1 between *PCSK9* and *APOB*. As we describe in the text, “human double knock-outs”, i.e. individuals with PTVs in both, *PCSK9* and *APOB*, show a reduction in plasma LDLc by 41.2 mg/dl compared to individuals with PTVs in only one of the two genes, and by 91.7 mg/dl compared to individuals with no PTV in either of the two genes. Strong evidence exists that lifelong genetic reduction of plasma LDLc translates well into efficient protection from cardiovascular disease, and that this effect is dose-dependent (e.g. Baigent et al., 2005; PMID: 16214597). Therefore, it can be assumed that therapeutics which inhibit *PCSK9* and *APOB* function similar to genetic inhibition will be efficacious, and that a combination therapy against both targets simultaneously is likely to lower LDLc and with this cardiovascular risk even further than a monotherapy against just one of these genes.

We hypothesize that gene pairs for which AEs can be detected through our analyses have a higher probability to yield successful combination therapies than those where no AEs can be detected at given sample sizes, which will need to be demonstrated in future studies. For instance, the AEs we identified and validated through both, genetics and coRNAi between *HMGCR* and *MLXIPL*, *NCAN* or *SIK3* propose that these little characterized genes could potentially be attractive new targets for LDLc lowering therapies on top of statins.

3) The authors use a model selection approach (with a fairly standard BIC implementation), which is essentially asking “if I had to guess what the genetic relationship is, what would my best guess be”. This approach might be fine for picking a working hypothesis, but it is a completely inadequate standard of evidence for claiming with any confidence that an interaction (or non-interaction) really exists. The authors should find some way of estimating confidence in their conclusions, e.g. showing that confidence intervals for the double-perturbation effect (e.g., obtained from PTV data by bootstrap resampling from subjects) shows a departure in the right direction from the relevant single gene effect.

We thank the reviewer for pointing out potential shortcomings of the BIC model selection approach. In our revised manuscript, we utilize BIC now only for discovery/hypothesis generation, but then follow-up with a more in depth statistical analysis of our results: robust linear model fitting complemented by correcting for multiple hypothesis testing using FDR correction. In our original manuscript we had already conducted such additional analyses, yet only for part of our coRNAi analyses. In our revised manuscript these additional analyses have now been extended to the full coRNAi dataset as well as to genetic analyses.

In brief, we performed FDR-correction as described by Benjamini and Hochberg, 1995. P-values were extracted from the t-values (generated by `rlm` fit) for effects of single genes and interaction terms. For both genetics and coRNAi, we set the threshold to call a gene pair as GI to an FDR-corrected p-value for the interaction term as $p < 0.005$, with FDR correction being based on the number of interactions between gene/variant pairs tested ($n=1,740$ for PTV-PTV interaction, $n=1,890$ for SNP-SNP interaction, $n=3,240$ for PTV-SNP interaction, $n=120$ for PTV-PRS interaction, and $n=435$ for coRNAi primary screen and $n=36$ for coRNAi validation screen) times the number of genetic effects tested ($n=3$, single-gene effects for gene1 and gene2, and the interaction effect if testable). FDR-correction was done separately for PTV-PTV, SNP-SNP, PTV-SNP, and PTV-PRS interaction for genetic analyses given the different nature of genetic variations tested, but a rather stringent significance cutoff was chosen at FDR corrected $p < 0.005$ for all analyses. For the effects of single genes and interaction terms we also calculated 95% confidence intervals provided in Supplementary Tables as an additional measurement illustrating significance of identified effects. Confidence intervals that assume a normal distribution for robust linear regression coefficients ($\text{Beta} \pm 1.96 \times [\text{standard error of the mean}]$) are now presented in Supplementary Tables.

We now describe this new analysis strategy in the revised Methods section (adjusted for each data type it was applied to) as such: *“In order to also estimate statistical significance of gene interaction or additive effect for each {PTV}, we calculated normal distribution approximated p-values from the t-statistics of the robust linear regression full model (Model 3 or Model 4 depending on the gene pair), including the interaction term (if interaction testable), as well as the {PTV} effect for each single gene, for all gene pairs and all four lipids measurements. To correct for multiple comparisons, the p-values were then adjusted using the false discovery rate (FDR) method⁵⁹, after which the gene interactions as well as additive effects were identified in a two-step manner: if the FDR-corrected p-value of the interaction term was less than 0.005 and the BIC indicated Model 4 as the best fitting model, then the gene pair was considered as being a significant GI; if the FDR-corrected p-value of the interaction term was larger than 0.01 or untestable, but FDR-corrected p-value for each single-gene PTV effect was less than 0.005 and the BIC indicated Model 3 as the best fitting model, then the gene pair was considered as a significant AE.”* We have also added the above description in a dedicated paragraph termed “FDR correction” and will be providing the R-scripts used for our calculations in a public repository (<https://git.embl.de/grp-almf/genetic-interactions-screen-lipid-levels>).

4) The authors only provide measures of statistical significance for the RNAi approach. This calculation is flawed, in that the authors seem to be (line 852) assuming that a t-statistic follows a normal distribution. Given that t distributions typically have fatter tails than normal distributions, this approach will provide misleadingly low measures of significance. Perhaps their FDR procedure accounts for this, but no details are given on the procedure (e.g., what was the source of null distribution for this calculation).

In our revised manuscript, we now provide the sample sizes on which our statistical analyses rely on. We have also added additional statistical analyses and introduced wording in the Methods that should better detail how FDR correction was performed (see also our above response to this Reviewer's Comment #3). As our revised Supplemental Tables show, our analyses are typically based on a few hundred to many thousand data points, depending on the respective approach (e.g. for coRNAi, see Supplementary Tables 10 and 11). At the given sample sizes, the t-distribution should be largely indistinguishable from a normal distribution for describing means of estimated effects so that potential differences should become negligible.

Violin and box blots in revised Figures 1, 3, 4 and Supplementary Fig.1 and 3, as well as Supplemental Tables should now provide a better impression of the distribution of plasma lipid levels in UK Biobank participants, as well as the distribution of cell-based fluorescent LDLc levels per image frame.

5) Interpretation of LDL uptake data should consider that fluorescence measures of Dil-LDL will include LDL bound to the cell surface (e.g., line 224-5 the higher Dil-LDL signal in LDLRAP1 knockdown may also be interpreted as more binding to LDLR on cell surface (makes sense since LDLRAP1 is required for positioning LDLR at clathrin-coated pits). The suggested positive GI between LDLR and LDLRAP1 seems unfounded. LDLRAP1 functions in the internalization of the LDLR-LDL complex, and it has been well established that absence of LDLRAP1 results in accumulation of LDLR at the cell surface. The authors use a fluorescent LDL molecule (Dil-LDL), which does not distinguish between cell-surface bound and internalized LDL molecules, therefore while this may be a buffering interaction in that loss of LDLR and LDLRAP1 leads to a similar phenotype as loss of LDLR, the results of the co-RNAi screen are perhaps being misinterpreted.

We appreciate the reviewer's comment. Our LDL-uptake assay includes a step that efficiently removes LDLc bound to the plasma membrane by a short acid wash as described in Methods (page 30) and in more details in our earlier publications applying an identical protocol, e.g. Bartz et al., 2009, PMID: 19583955; Blattmann et al., 2013, PMID: 23468663; or Thormaehlen et al., 2015, PMID: 25627241.

We have also conducted additional experiments to substantiate that under our experimental settings LDLR does not accumulate at the cell surface upon knockdown of LDLRAP1. These new data are now included as Supplementary Fig. 5 and described in the main text (page 8) as: "*Conversely, knockdown of LDLR strongly inhibited, whereas knockdown of LDLRAP1 increased cellular LDLc-uptake under our experimental conditions, an effect that could not be explained by elevated LDLR levels at the plasma membrane (Supplementary Fig. 5).*"

We agree that an *increased* LDLc-uptake upon LDLRAP1 knockdown may at first view seem counterintuitive since homozygous loss-of-function of *LDLRAP1* is a cause for Familial Hypercholestermia (FH), characterized by *reduced* LDLc plasma clearance. However, a modestly increased Dil-LDL uptake upon treating HeLa cells with LDLRAP1 siRNA is highly reproducible. For instance, four out of five *LDLRAP1* siRNAs tested in our earlier study (Blattmann et al., 2013, PMID: 23468663), including the one selected for this current manuscript, resulted in increased Dil-LDL uptake (although not all meeting significance thresholds) and three siRNAs increased free cholesterol levels in cells (of which two met our stringent significance criteria). We had interpreted these results by the fact that first, siRNA knockdown does not lead to an entire abrogation of cellular LDLRAP1 levels (see also Supplementary Table 2) as in autosomal-recessive FH. Second, LDLR internalization into cells occurs through at least two partially redundant and interdependent pathways, one of which involves LDLRAP1, the other DAB2 (disabled-2). Unlike in hepatocytes, in the HeLa cells used in our screening experiments *DAB2* is expressed at high levels (unpublished observation) and it is reported to be a key factor for efficient LDLR clustering into clathrin-coated pits (CCPs) and LDL endocytosis (Maurer and Cooper, 2006, PMID: 16984970). We hypothesize that DAB2-dependent LDLR internalization might be a compensating mechanism for LDLc internalization upon partial knockdown of *LDLRAP1*, potentially explaining a modestly increased LDLc uptake in this cell line, although we consider it as beyond the scope of this present manuscript to investigate this further.

Under the scenario of joint knockdown of *LDLR* and *LDLRAP1*, absence of the receptor is a limiting step for the endocytosis of LDLc, thus we observe a *positive alleviating GI* (Figure 3c,e), consistent with both proteins acting interdependently in the process of LDLR-internalization (see also Reviewer Figure 1 in our response to Reviewer #3, Comment 4).

6) Lines 188-191: The p.S447Ter stop-gain mutation in LPL is the only stop-gain variant in LPL known to have a gain-of-function effect (see PMIDs 1752947; 16574898) . As such, it would likely not be expected for other deleterious stop-gains in LPL to mitigate a high polygenic risk for elevated TG. On the contrary, loss of LPL function would be expected to have the opposite effect.

In our revised manuscript we have now added sensitivity analyses and calculated the contribution of *LPL* to the featured genetic effects not only using overall PTV-burden, but also adjusted for p.S447X. As we had hypothesized already earlier, these new data show that p.S447X is indeed the key driver for

the AE between *LPL* with *APOB*, *LDLR* and *PCSK9*. The comparative analyses are now provided as new Supplementary Table 6 and referred to in the text as: “Conditioning these gene pairs for the most prevalent *LPL* PTV, *p.S447X* abrogated the signals, suggesting the AEs with *LPL* were primarily driven through this distinct gain-of-function allele instead of a loss-of-function mechanism (Supplementary Table 6).” (page 5)

Likewise, *p.S447X* is also the main driver of the GI identified between *LPL* and TG-PRS. We highlight this in the text as such: “Of all combinations tested, only PTV-burden in *LPL* showed evidence for a GI with the PRS for TG ($p=1.77 \times 10^{-14}$; $\beta=-0.03$) (Figure 2f; Supplementary Table 9). The GI remained significant ($p < 2 \times 10^{-16}$, $\beta=-0.04$) in a sensitivity analysis using the *p.S447X* variant alone instead of *LPL* PTV-burden, proposing an alleviating GI of *p.S447X* variant on TG-PRS. These results are consistent with the hypothesis that a high polygenic risk for elevated TG can be mitigated by a concomitant gain-of-function mutation in *LPL*.” (page 7)

7) There should be discussion (both in general and specifically for each relevant finding) of the issue that GWAS SNPs cannot generally be unambiguously mapped to a single gene, so that the association may stem from a causal variant from another gene that is in LD with variants in the specific genes discussed.

We certainly recognize the challenges to relate from GWAS finding to underlying causal gene and had highlighted already in our original manuscript as a limitation that by using genetic tools (such as fine mapping, colocalization and Mendelian Randomization) we and others have established probable causality for only 5 of the 30 candidate genes (3 via liver eQTLs, 2 via plasma pQTLs; now discussed on pages 13-14). Indeed, as we emphasize in our manuscript, the selection of candidate genes was primarily based on prior insights on their respective biological functions (15 genes), proximity of the lead SNP to the candidate gene, as well as the fact that we had identified and validated 28 of the genes as impacting cellular DiI-LDL uptake and/or cholesterol levels in our prior study (Blattmann et al., 2013, PMID: 23468663). In that earlier study we had systematically tested the function of 133 genes in 56 lipid and CAD GWAS loci, and the selected genes had scored most prominently. While certainly not unambiguous, we feel they can in most instances at least be seen as a good proxy for the likely causal genes. We detail our gene and lead SNP selection criteria both in Methods, as well as the first paragraph of Results entitled “Study outline”.

In our revised manuscript, we further emphasize these limitations on page 6 as “We next tested for pairwise genetic effects using 28 lipid/CAD GWAS lead SNPs representing the 30 loci in 378,033 unrelated individuals of European ancestry in the UK Biobank²⁵ as proxies for the respective candidate genes.” We have also added the following sentence to the discussion: “Conversely, the large numbers of individually low-impact GWAS lead SNPs may not unambiguously be mapped to a respective gene, so that observed associations may stem from a causal variant from another gene in LD.” (page 13-14)

Minor issues

8) Re: exome data analysis, it was not made clear if truncations appearing late in the coding region (which are often not damaging) were excluded. In addition to PTVs, the authors might have also considered missense variants that were confidently predicted as damaging by a computational predictor (although this would require some calibration against a sufficient number of known pathogenic and benign variants not used in training the predictor, which may not be available for all 30 genes of interest).

Yes, PTVs appearing late in the coding region were excluded from our analysis. We have now specified this in Methods as such: “PTVs were called from a Regeneron QC-passing “Goldilocks” set of genetic variants using Variant Effect Predictor v96²⁸ and the LOFTEE plugin²⁷. LOFTEE applies a range of filters on stop-gained, splice-site disrupting and frameshift variants in order to exclude putative PTVs due to variant annotation and sequencing mapping errors that are unlikely to significantly disrupt gene function. For instance, stop-gained and frameshift variants that are within 50 kb of the end of the transcript will be flagged as “low-confidence”. For our analysis, we only considered variants predicted as PTVs and flagged as “high confidence” by LOFTEE for each canonical transcript (as defined in Ensembl).” (page 23)

We agree that extending our analyses to missense variants may be potentially very interesting, but as recognized by the Reviewer and e.g. demonstrated in an earlier manuscript of ours at the case of *LDLR* (Thormaehlen et al., 2015, PMID: 25627241) the challenges for annotating missense variants as unambiguously “damaging” remain substantial even in well-established disease genes so that we think such analysis is better suited for a future study.

9) Lines 141-147: Should clarify what is the reduction in cholesterol due to the additive genetic interaction between PCSK9 and APOB. What baseline is the reduction of 40.01 mg/dl from, the average level of single gene carriers, this is not clear.

We have now clarified the extent of absolute LDLc reduction as such:

“For instance, while control individuals without PTVs in either PCSK9 or APOB had average levels of 138.3 mg/dl LDLc, PTVs in PCSK9 and APOB individually reduced mean plasma LDLc by 34.5 mg/dl and 69.1 mg/dl relative to individuals without PTVs in these genes, consistent with previous reports³⁰⁻³¹. However, the three UK Biobank participants (“human double knock-outs”) who carried both, PCSK9 and APOB PTVs, showed on average a further reduction in plasma LDLc by 41.2 mg/dl compared to individuals with PTVs in only one of the two genes, and by 91.7 mg/dl compared to individuals with no PTV in either of the two genes (Figure 1b, Supplementary Fig. 1a), suggesting considerable additional protection from CAD.” (page 5)

10) Lines 165: The authors mention the SNP rs4420638, which is located in APOC1, a gene highly implicated in CHD. The SNP is in linkage disequilibrium with SNPs in APOE. If there are suggested cis-effects between this SNP and the locus in question, the biology of this interaction should be discussed, or at minimum the causal gene mentioned, as this SNP is associated with increased TC, LDL and TG levels.

We thank the reviewer for pointing out that this SNP is located in *APOC1* as one of the genes in the discussed 19q13.32 gene cluster which also includes additional genes to the ones mentioned in our earlier manuscript. In the revised manuscript we have now re-phrased this sentence as: *“The strongest driver of interactions came from the 19q13.32 APOE locus encompassing a cluster of genes that was contributing to 20 of the 25 GIs identified across all traits.”* In our prior publication (Blattmann et al., 2013, PMID: 23468663) we had studied the impact of knocking down also *APOC1* as one of 11 genes at this locus, but since in that study only one out of six *APOC1* siRNAs studied had shown a significant effect under our experimental conditions, *APOC1* failed to meet our inclusion criteria for this current study.

11) Lines 342-344: It is unclear how you would expect to detect the effects of a protein functioning in the trans-endothelial migration of leukocytes in this co-RNAi screen examining LDL-uptake (or surface-binding).

Since the previously described *PVRL2* genetic effects failed to meet our adjusted and more stringent statistical criteria of measuring an impact on cellular LDL-uptake, this gene and its involvement in trans-endothelial migration of leukocytes is no longer discussed in the main text of our revised manuscript.

12) Lines 48-50: “For the vast majority of dyslipidemic individuals, however, no single-gene mutation can be identified.” Should clarify that monogenic single variants could have been missed for many reasons, e.g., coding VUS or non-obvious splice-altering or other non-coding variants.

In the revised manuscript we have changed this sentence to *“For the vast majority of dyslipidemic individuals, however, no single-gene mutation can be identified or remain undetected without substantial follow-up.”* (page 2)

13) Figure 1. b & c: Statistical tests needed. Should also include plot for interaction between LPL-APOB on TG levels.

Statistical test results have now been added to Figures 1b-e. For all four gene pairs featured in Figure 1, distribution and statistical test results for the remaining lipid parameters are now visualized in the new Supplementary Fig. 1, and detailed background information on statistics for each gene pair tested is provided in Supplementary Tables.

14) Lines 42-43: As worded, implies that only heritability influences blood lipids and CAD and neglects known environmental influences

The first sentence of the Introduction now reads as: “*Genome-wide association studies (GWAS) have firmly established that a substantial fraction of variation in blood lipid levels and the risk of coronary artery disease (CAD) is heritable.”*

15) Lines 68-69: Should cite substantial earlier literature that GIs can assist in nominating drug pair therapeutics.

We now cite four additional papers that propose that GIs can assist in nominating drug pair therapeutics:

Han et al. *Synergistic drug combinations for cancer identified in a CRISPR screen for pairwise genetic interactions* (Nature Biotechnology 2017; PMID: 28319085)

Shen et al. *Combinatorial CRISPR-Cas9 screens for de novo mapping of genetic interactions* (Nature Methods 2017; PMID: 28319113)

Jo et al. *Yeast genetic interaction screen of human genes associated with amyotrophic lateral sclerosis: identification of MAP2K5 kinase as a potential drug target.* (Genome Research 2017; PMID: 28596290)

Cokol, M. et al. *Systematic exploration of synergistic drug pairs.* (Molecular Systems Biology 2011; PMID: 22068327)

16) Box and whisker plots don't do a good job of conveying subtle differences in the position or shape of distributions. Authors should consider representing/comparing distributions as histograms.

Following the recommendation of the Reviewer, we have changed the plots in Figure 1 and Supplementary Fig. 1 to violin plots.

Reviewer #2 (Remarks to the Author):

Overall:

Zimon M and Runz H and colleagues evaluate genetic interactions between candidate lipid-associated alleles and compare to a combined RNA interference assay. They are motivated to discover drug targets for lipids that may be particularly effective in combination. Human genetic analyses occur in the UK Biobank with privileged access to 240K whole exome sequences as well as available GWAS data. They observe that much of the effects are independent/additive with few interactions. Two interactions appear to be consistent in RNA interference experiments. I have several concerns regarding the methods, and particularly the claims in the present study.

Major:

1. The design of this study is motivated by identifying gene interactions for known lipid genes to discover optimal drug combinations. Each of these genes are previously known to contribute to lipid variation independently, and this additively, already. The authors cite that statins as well as other cholesterol-lowering medicines lower LDL-C levels and CAD risk beyond statin

treatment alone as a motivation – that’s simply because these are independent targets. This is not a property of a uniquely synergistic effect. Therefore, the “additive” gene interactions identified are of unclear significance. I also worry that using the phrasing “gene interactions” actually implies a significant interaction (i.e., non-additive effect) and may be misleading. It may be best only describe non-additive interactions in this paper and remove additive effects- this latter aspect merely leads to confusion and is not novel.

We thank the reviewer for pointing out that the terminology we used in the original version of our manuscript may have been ambiguous. Indeed, as we detail also in our response to Reviewer #1, Comment 1, “additive genetic interactions (aGIs)” were supposed to signify pairwise effects that are the sum of the individual genes’ effects. To avoid confusion, our revised manuscript now refers to “aGIs” as “additive effects (AEs)” while the term “genetic interaction (GI)” remains reserved for non-additive effects.

We believe that our study considerably benefits from an interrogation of both, AEs and GIs for several reasons: First, our results support earlier studies that - at least for lipid traits and a set of highly probable GWAS candidate genes - AEs by far outnumber GIs. Second, we extend these earlier insights, which were largely derived from GWAS loci, now to pairs of genes through leveraging PTVs as well as to functional insights gained through cell-based coRNAi. Third, we consider AEs as highly informative, especially - as we outline in the text - for decisions which among many possible gene pairs to prioritize as putative drug targets for combination therapies. This is nicely exemplified at the case of *PCSK9* and *APOB*: Both genes are the targets of approved lipid-lowering drugs. However, to the best of our knowledge our study is the first to describe “human double knock-outs” in whom blood LDLc levels are further reduced than in “single knock-outs”. Such “experiment of nature” at the population level thus strongly supports testing drugs against both targets in combination, since this can be expected to most probably reduce CAD risk further. Fourth, additional AEs validated in our study through both genetics and coRNAi encompass less well-characterized genes that may stimulate experimental follow-up for finding new lipids lowering treatments that go beyond standard-of-care. Examples for these include AEs between *HMGCR* and *MLXIPL*, *NCAN* or *SIK3*, introducing these little characterized genes as potentially attractive new targets for LDLc lowering therapies on top of statins. Last, it is tempting to speculate that once sample sizes for genetics will reach large enough numbers (eventually many millions, as recently proposed by Hivert et al., 2021; PMID: 33811805), non-additive effects may become evident for some effects identified here as AEs. Candidates for this might include *APOB-HMGCR* and *PCSK9-TMEM57* which genetics nominates as AEs, but coRNAi, which is able to test for more dynamic effects rather than lifelong effects conferred through variation in the germline, proposes as putative LDLc-uptake simulating GIs.

2. A major limitation of the present work is the restriction just to 30 known lipid genes that are known to independently contribute to lipid variation. Therefore, the likelihood for identifying non-additive interactions is low. Extending beyond these genes and potentially using known protein-protein interaction data is more likely to yield novel insights.

We agree with the Reviewer that bioinformatic exploration of protein-protein interaction data could have been another interesting starting point to nominate gene pairs for investigating genetic effects that could be explored in future studies. Alternatively, an entirely hypothesis-free, genome-scale interaction analysis by both genetics and cell-based function would have also been a desirable approach. However, current sample sizes and functional platforms, as well the penalty for multiple-test correction, when conducting interaction analyses at genome scale, would have drowned any eventual GI signals. Instead, we chose to follow a “closest to hypothesis-free approach” by zooming in on a set of candidate genes for which two (largely) hypothesis-free approaches converged: lipid GWAS and our previous functional screen for cell-based LDL-uptake modulating genes (Blattmann et al., 2013; PMID: 23468663). In our revised manuscript we now better explain our motivation for candidate gene selection (see also our response to this Reviewer’s Comment #6). We do not think that our approach of choosing candidate genes primarily from GWAS loci would introduce substantial biases against the detection of GIs (which consistent with our results are likely to be rare also when following alternative strategies).

3. The combinatorial RNA interference readout only measures LDL uptake into cultured cells. This is only one aspect of LDL cholesterol regulation. Others not assayed include apoB-

containing lipoprotein synthesis in the liver, VLDL to LDL conversion via LPL and HL, transfer of cholesterol between HDL and LDL via CETP, etc. This substantially hinders the ability to identify new insights even in the candidate list of 30 known independent lipid genes.

We agree that measuring LDLc uptake reflects only one out of the multiple complex biological mechanisms necessary to maintain plasma and cellular lipid levels and highlight this as a potential limitation in our Discussion now as such: “*First, our functional analyses were limited to measuring LDLc uptake into cells, which reflects a relevant, yet only a partial aspect of the many possible mechanisms by which a gene can modulate plasma lipid levels.*” (page 13) However, given the central therapeutic relevance of LDLc lowering to treat or prevent cardiovascular disease and the fact that several established therapeutics including statins leverage LDLc uptake modulation as a highly efficient way to lower LDLc, it is a central biological process to monitor when studying lipid biology. Also, to the best of our knowledge no study has yet systematically tested the relevance of pairwise genetic effects on cellular LDLc uptake. Monitoring additional lipid regulatory pathways with functional assays that can reliably measure genetic effects at scale would certainly be interesting, but we feel such sets of experiments go substantially beyond the scope of this current manuscript and will have to await future studies.

4. Could non-additive interactions be driven by cryptic relatedness? Incorporating random effects of the genotypes would be helpful. Or, more simply, assessing correlation for the variants assessed would be helpful to address this concern.

As detailed in Methods, we have applied stringent measures to correct for the risk of bias from cryptic relatedness. For instance, we excluded from our analyses one member (at random) from each pair of 2nd degree relatives. We also used the public UK Biobank genotype data to conduct principal component analyses that control for ancestry and excluded individuals with non-white British ancestry from our main analysis, further reducing the risk that findings in our manuscript could be driven by relatedness.

5. Could non-additive interactions be driven by selection? If two independent alleles separately contribute to a deleterious phenotype (such as increased LDL-C), if they happen to co-occur in the same individual who is a healthy UK Biobank phenotype, this could lead to an apparently less severe phenotype in this individual than expected.

We agree that selection is likely a non-negligible factor for the presence or absence of genetic effects. There is strong evidence that due to their high functional impact PTVs are under a high selective pressure in the human germline, which is reflected by their relative rarity. Consistently, we identify substantially fewer AEs or GIs for PTVs than for low-impact, yet prevalent GWAS lead SNPs. For instance, for LDLc only four gene pairs showed AEs upon PTV-PTV testing as opposed to 142 upon SNP-SNP testing. For those few individuals who carry PTVs in both genes the phenotype may indeed be less severe than expected for single-PTV or non-carriers, best reflected at the case of the three UK Biobank participants who carry PTVs in both, *PCSK9* and *APOB* and show substantially lower LDLc levels (Figure 1c; see also our response to Reviewer #1, Comment 9). Identifying GIs and even most AEs through PTV-PTV analyses, and with this robust conclusions as to their evolution and selection is likely to require substantially larger sample sizes as currently available, at least for lipid levels that typically lead to disease manifestation only at later stages in life.

6. I'm not clear how common variant SNPs were selected. The methods indicate: “Twenty-eight lead SNPs mapped to the 30 lipid GWAS genes were extracted...” How were these connected biologically with each of the 30 lipid GWAS genes? Many of these loci also have locus heterogeneity but it appears this was not considered.

The literature sources for the named lipid GWAS lead SNPs are provided in Supplementary Table 1, column I. For all lipid loci analyzed we queried whether more strongly associated SNPs had been reported since the original publication using PhenoScanner vs2 (<http://www.phenoscanner.medschl.cam.ac.uk>). Where this was the case, the more recent sentinel SNP was selected for analysis. When a respective locus was associated with multiple lipid phenotypes,

the SNP with the lowest reported p-value association with LDLc was chosen to be the lead SNP. We describe the SNP selection process now more thoroughly in Methods.

An direct genetic link between a lead SNP and a respective candidate gene could only be established in few instances, e.g. for two genes where the reported SNPs were eQTLs that colocalized with target gene liver expression based on GTEx, or for three genes for which the selected SNPs had been identified to be cis-pQTLs in plasma (see Supplementary Table 2). A lead SNP at the *SORT1/CELSR2* locus (rs629301) showed a positive colocalization signal, but the *cis*-eQTL co-localized with both genes, so SNP-based GIs for these genes could not be analysed separately. However, for all genes analyzed there was prior biological evidence for a role in lipid regulation, either through the literature (15 genes), or based on our earlier study by Blattmann et al., 2013 (PMID: 23468663) in which we had studied how single gene knockdown of 133 candidate genes in 56 lipid loci impacted LDLc uptake into cultured cells. In that prior study, for 38 of the 56 loci we analyzed all protein-coding genes within ± 50 kb of the respective reported lead SNPs. The 18 remaining loci were represented by candidate genes closest to the lead SNPs (as detailed in the Methods and Supplementary Tables 1 and 2 of Blattmann et al., 2013). 28 of the 30 genes studied in the current manuscript had scored as functional regulators of LDLc uptake, cellular levels of free cholesterol, and/or LDL-receptor (LDLR) mRNA or protein levels. Genes from multigenic GWAS loci that did not individually score as modulating LDLc uptake and/or cellular lipid levels upon siRNA knockdown were not analyzed in the current study.

7. 4 models for several pairwise interaction tests for 4 traits each pairing are described throughout the methods but criteria for statistical significance accounting for multiple hypothesis testing are not described in the methods. This comment applies to both the human genetic analyses as well as the RNAi analysis. As such, I'm not quite clear which of the present analyses are truly statistical significant if any based on the interaction p-values.

In our revised manuscript, multiple hypothesis testing has now been conducted for both, genetic as well as coRNAi analyses (see also our response to Reviewer #1, Comment 3). Also all other statistical analyses have been harmonized between genetics and coRNAi. In brief, for all datatypes we first conducted Schwarz's Bayesian Information Criterion (BIC) scoring to generate a hypothesis about the best model to explain the data and goodness of fit. This was followed by robust linear model fitting and correcting for multiple hypothesis testing using FDR (according to Benjamini and Hochberg, 1995). For both genetics and coRNAi, we set the threshold to call a gene pair as GI to an FDR-corrected p-value for the interaction term as $p < 0.005$, with FDR correction being based on the number of interactions between gene/variant pairs tested ($n=1,740$ for PTV-PTV interaction, $n=1,890$ for SNP-SNP interaction, $n=3,240$ for PTV-SNP interaction, $n=120$ for PTV-PRS interaction, $n=435$ for coRNAi primary screen and $n=36$ for coRNAi validation screen) times the number of genetic effects tested ($n=3$, single-gene effects for gene1 and gene2, and the interaction effect if testable). FDR-correction was done separately for PTV-PTV, SNP-SNP, PTV-SNP, and PTV-PRS interaction for genetic analyses given the different nature of genetic variations, but a rather stringent significance cutoff was chosen at FDR corrected $p < 0.005$ for all analyses apart from coRNAi validation experiments for which we chose an FDR corrected $p < 0.01$.

We are now detailing our statistical analysis strategy for all datasets and approaches in Methods and have introduced an additional paragraph entitled "FDR correction" (page 21) to further clarify why we consider the reported findings as truly statistically significant. We now also provide access to the well-described R-code which we used for analyses of genetic and co-RNAi data at a dedicated public GitLab site that can be accessed via: <https://git.embl.de/grp-almf/genetic-interactions-screen-lipid-levels>.

8. In the PTV gene-burden interaction analyses, why are LPL-APOB or PCSK9-APOB considered significant interactions? P-values for non-additive interactions for 3 of the 4 tests are > 0.05 , and the 4th appears to not satisfy Bonferroni correction. I believe this claim is for additive effects. Showing independent effects for these is not novel and is confusing when using the term "interaction" when the p-values for interaction are nonsignificant. It's not surprising that APOB PTVs + PCSK9 PTVs lead to reduced cholesterol in an additive fashion and is likely not uniquely informative for a drug development program. Please refer to major point #1.

Indeed, already in our original manuscript we had classified the effects between *LPL-APOB* and *PCSK9-APOB* as additive. We hope that the revised terminology as detailed in our response to Reviewer #1, Comment 1 as well as this Reviewer's Comment 1 helps to further clarify this. In our revised manuscript using additional statistical approaches and a larger discovery cohort of 302,331 UK Biobank exomes, AEs for both gene pairs were confirmed (*PCSK9-APOB* for LDLc and TC; *LPL-APOB* for HDLc, LDLc and TG) and further AEs identified for two additional gene pairs (*LDLR-LPL* for LDLc; and *PCSK9-LPL* for LDLc; see revised Figure 1 and new Supplementary Fig. 1). While AEs between *APOB* and *PCSK9* may not necessarily be surprising based on what we know about the biology of these genes, to the best of our knowledge no "human double knockouts" have yet been described through a population-level exome study. We consider such demonstration of efficacy through an "experiment of nature" as highly informative for drug development, and since approved drugs exist against these targets eventually even of immediate interest for improving medical care.

9. A number of the SNP-SNP interactions identified are in very close genomic proximity (e.g. rs2228603-rs58542926, rs118147862- rs4420638, rs118147862- rs2075650, etc). Is this a byproduct of LD and not reflective of true biological interaction? This is particularly true at the APOE locus where the authors claim there are several non-additive genetic interactions.

We agree that a potential "bleed-through" effect of the massive *APOE* signal on these "cis-GIs" is difficult to differentiate from true genetic effects. This is one reason why we have highlighted when a putative GI involves Chr. 19q13.32 in an extra column in Table 1. Nevertheless, only a single of the twelve 'cis-GI' SNP pairs showed strong LD ($R^2=0.764$) (*NCAN-TM6SF2*) and two others weak LD ($R^2>0.1$; *ZNF259-SIK3*, *ZNF259-PAFAH1B2*), while no evidence for LD was found for the other presumed cis-GIs ($R^2<0.01$). This information has now been added to the Results (page 6).

10. Across all the human genetic interaction testing and coRNAi interaction testing, only two non-additive interactions were consistent in both sets of experiments (SORT1-TOMM40 and NCAN-TOMM40). Identification of these may be novel. But given all the various testing, I'm not sure that these survive Bonferroni correction. If they do not, additional replication data would be helpful.

CoRNAi analyses nominated the genetic effects between *TOMM40-SORT1* and *TOMM40-NCAN* as GIs in both, primary screen and validation analyses. The FDR-corrected p-values, were calculated as described by Benjamini and Hochberg, 1995 (see our response to this Reviewer's Comment #7), by pulling all p-values from single effects and the interaction term (3x435 p-values for primary and 3x36 p-values for validation coRNAi screen) and performing FDR correction on this set of p-values. Thus FDR-corrected p-values for *NCAN-TOMM40* were $p=3.23E-03$ for the primary and $p=3.45E-08$ for the validation screen, and for *SORT1-TOMM40* $p=9.95E-04$ and $p<e-15$, respectively (see also Table 2). For LDLc SNP-SNP analyses, we approximated *SORT1* by rs629301 (which as we state in our response to this Reviewer's Comment 6 may not unambiguously distinguish between *SORT1* and *CELSR2*), *NCAN* by rs2228603 and *TOMM40* by rs2075650 (which as we state in our response to this Reviewer's Comment 9 cannot be unambiguously differentiated from variants impacting *APOE*). For genetic analyses, too, BIC predicted GIs (model 4) as the most likely model and FDR-corrected p-values (with FDR correction performed on 3x1,890 p-values) were <0.005 ($p=1.11E-03$ for *SORT1-TOMM40* and $p=1.41E-11$ for *NCAN-TOMM40*) (see also Table 1). We did not introduce an additional correction when overlaying coRNAi and SNP-SNP interaction analyses since the two approaches are largely independent and the gene pairs going into this analysis were only few (e.g., 3 from LDLc SNP-SNP analysis). With the substantial functional and genetic replication/validation analyses conducted we thus consider these two gene/variant pairs as surviving multiple testing correction and thus most likely interacting as GIs.

11. SORT1-TOMM40 and NCAN-TOMM40 are the prioritized interactions as noted above but significance is unclear. These are "buffering" interactions, meaning it would be inefficient to therapeutically target both together?

This is correct, coRNAi proposes the GIs between *SORT1-TOMM40* and *NCAN-TOMM40* indeed as "buffering", or according to our revised terminology "*positive alleviating*". Note that since for SNP-SNP

based interaction analyses the effect sizes were only small we abstained from further sub-categorizing GIs. If cell-based coRNAi results translated 1:1 to therapies, it would indeed indicate that joint inhibition of both genes in the respective gene pair is unlikely to yield an additive therapeutic benefit beyond single gene modulation, and joint modulation might even be counterproductive. As such, these might be good examples for drug targets that should better *not* be pursued combination therapies. For drug discovery this is often even more valuable information since a waste of resources often outweighs missed opportunities. However, since GI directionality is not validated by genetics, we decided not to discuss this in the manuscript.

12. A paragraph of study limitations should be included in the Discussion.

The following paragraph now describes limitations of our study in the Discussion: *“Interestingly, a significant number of GIs identified through genetics and coRNAi in our study do not yet overlap. This may be explained by several reasons: First, our functional analyses were limited to measuring LDLc uptake into cells, which reflects a relevant, yet only a partial aspect of the many possible mechanisms by which a gene can modulate plasma lipid levels. Second, siRNA-based gene knockdown captures acute and rather severe functional effects, which may differ from the chronic and often compensated consequences upon lifelong modulation of a gene’s function through genetic variation. Third, despite the large number of samples used for genetics-based GI testing, the number of informative high-impact variants in the human germline may still be too discrete to comprehensively identify GIs. Conversely, the large numbers of individually low-impact GWAS lead SNPs may not unambiguously be mapped to a respective gene, so that observed associations may stem from a causal variant from another gene in LD. Regardless, the availability and rapid development of advanced high-throughput microscopy technology joint with the constantly increasing cohort sizes for genetic analyses will allow up-scaling of the approach taken here in future studies and with a high probability identify and validate further genetic effects.”*

Minor:

13. Page 1, line 31: “...not always translate to humans” should be “...do not always translate to humans.”

We have modified the sentence accordingly.

14. Several words are combined with hyphens. This appears unnecessary in several instances. e.g., “CAD-risk,” “statin-treatment,” “exome-sequencing,” “RNA-interference,” “LDL-uptake,” “QC-ed.”

We have removed all unnecessary hyphens throughout the manuscript.

15. “LDL” is used for low-density lipoprotein cholesterol and “HDL” for high-density lipoprotein cholesterol. Better terms would be “LDL-C,” “LDLc,” “LDL cholesterol,” etc or “HDL-C,” “HDLc,” “HDL cholesterol,” etc, respectively.

We now utilize the terms as proposed by the Reviewer, “LDL” is now “LDLc”, “HDL” is now “HDLc”.

16. The first introduction paragraph regarding monogenic and polygenic basis of lipids among dyslipidemic individuals currently reads as unrelated to the scientific premise of the present study. I would consider starting with the motivation of the study up front for better orientation.

We thank the reviewer for this suggestion, but felt it was useful to discuss our pairwise approach to testing genetic effects in the context of polygenic risk scores (PRS) and the increasing evidence for (at least) additive effects between PRS and monogenic diseases. Since we do conduct an analysis for effects between candidate gene PTV-burden and lipid PRS in our manuscript (see e.g. Figure 4f), we felt it appropriate to provide the necessary background in the introduction.

17. In the methods, on page 23, there is a sentence “In total, 30,125 CAD cases identified and the cohort was adjusted for age, sex, smoking status, alcohol drinking status, BMI and lipid medication use.” What was adjusted? What is the outcome and what’s the exposure? Were principal components considered?

Our CAD analysis was conducted using CAD as the outcome and lead SNPs for lipids/CAD loci as exposures assessed in a pair-wise manner. We adjusted for age, sex, smoking status, alcohol drinking status, BMI, lipid medication use, as well as top ten principle components as covariates. This is now provided in more detail in the Methods (page 22).

18. “Synergistic” and “buffering” should be defined in the Methods.

As detailed in our response to Reviewer #1, Comment 1, we have introduced new terminology in our revised manuscript. “Synergistic GIs” are now called “*negative aggravating GIs*” and “buffering GIs” are now called “*positive alleviating or suppressive GIs*”. This terminology is based on an article by Benjamin Boucher and Sarah Jenna (2013; PMID: 24381582) and also recommended terminology by EMBL-EBI ontology search:

https://www.ebi.ac.uk/ols/ontologies/mi/terms?iri=http%3A%2F%2Fpurl.obolibrary.org%2Fobo%2FMI_0208&viewMode=All&siblings=false.

We introduce the new terminology in the Results section, page 4 as such: “*For each gene pair, both the additive effects (AEs) (model 3), defined by the sum of effects from each gene or variant individually, as well as the genetic interactions (GIs) (model 4), represented by observed effects different than the expected additive effect, were calculated, with GIs being divided further into negative (aggravating) or positive (either alleviating or suppressive) (Figure 1a and Methods)*”. We have further modified Figure 1a accordingly and also define this terminology in Methods.

19. Page 5 line 119-120. The authors indicate “we sequenced the exomes of 200,654 UK Biobank participants...” If the authors are not going to be describing the exome sequencing methods, then better phrasing might be “we obtained the exome sequences of 200,654 UK Biobank participants...”

Our revised manuscript utilizes a further enlarged release of UK Biobank exomes and now reads as suggested: “*We obtained the quality-controlled exome sequences of 302,331 UK Biobank participants, annotated PTVs using Variant Effect Predictor v96²⁸ and the LOFTEE plugin²⁷, and identified 573,369 high-confidence PTVs in the canonical transcripts of 19,076 genes.*” (page 4)

20. On page 5 line 123 – page 6 line 127. Most PTVs in LDLR, PCSK9, APOB have a phenotype consistent with familial hypercholesterolemia. This is likely the case for LDLR. But most PCSK9 or APOB PTVs have reduced lipids and that’s why most of these are not listed as pathogenic in Clinvar.

We thank the Reviewer for this observation, this formulation has been an oversight. Indeed, consistent with the literature and as evident from Supplementary Table S4, PTVs in *PCSK9* and *APOB* are associated with reduced LDLc, while PTVs in *LDLR* are associated with increased LDLc. The revised text now reads as: “*For instance, we discovered 41 different PTVs in LDLR, 57 in PCSK9 and 142 in APOB. Most PTVs in these three genes co-occurred with strongly abnormal plasma LDLc levels in heterozygote carriers, with 45 PTVs annotated as pathogenic or likely pathogenic in ClinVar²⁹.*”

Reviewer #3 (Remarks to the Author):

The authors looked for genetic interactions in blood lipid and coronary artery disease (CAD) traits in a large cohort, by mainly focusing on 30 significant GWAS hits that were previously functionally validated by the same group using RNAi knock down. Using exon sequencing, RNAi

knock-down and available genome sequencing data, they sought for rare-common variants interactions (PTV-SNP), common-common variants interactions (SNP-SNP), and interactions between pairwise RNAi knock downs. In summary, they found that these traits related to lipid metabolism is mainly driven by additive effect, and non-additive genetic interactions are rare and only contribute marginally to the traits.

Comments:

1. The definition of “additive genetic interaction (aGI)” is misleading. What the authors testing here is the additive genetic affect, not a GI, as they defined themselves in figure 1. The authors should reword this in their manuscript.

We thank the Reviewer for this guidance on nomenclature. As detailed also in our responses to Reviewer #1, Comment 1 and Reviewer #2, Comment 1, we have now updated the nomenclature and in our revised manuscript now use the term “*additive effects*” or “*AEs*” as suggested. Likewise, effects previously described as “*non-additive GIs*” are now referred to as “genetic interactions (GIs)”.

We introduce the new terminology in the Results section, page 4 as such: “*For each gene pair, both the additive effects (AEs) (model 3), defined by the sum of effects from each gene or variant individually, as well as the genetic interactions (GIs) (model 4), represented by observed effects different than the expected additive effect, were calculated, with GIs being divided further into negative (aggravating) or positive (either alleviating or suppressive) (Figure 1a and Methods)*”. We have further modified Figure 1a according to this terminology and also adjusted the title of our manuscript which now reads: “Pairwise genetic effects modulate lipid plasma levels and cellular uptake”. The revised terminology is being used throughout the text, figures and tables.

2. Related to the first point, all results and analyses regarding “aGI” are just additive genetic effects. The only results that are relevant to genetic interactions are the “naGI” data as defined by the authors.

Indeed, “aGIs” were supposed to reflect “additive effects”, and we hope our revised terminology which now follows EMBL-EBI ontology (see also our response to Reviewer #1, Comment 1) is now less ambiguous. As we detail in our responses to Reviewer #1, Comment 2 and Reviewer #2, Comment 1, not only the identification of GIs, but also of “just” AEs at population scale can yield insights into biological mechanisms and help in prioritizing drug target pairs for combination therapies, especially when analyses are conducted with coding variants or tools that directly address gene products as in our study.

3. When talking about genetic interactions (naGIs), I don’t have a clear sense of the magnitude of beta related to interactions, vs beta related to additive effects. I think it would be helpful for the authors to present this information in a more straightforward manner.

We thank the Reviewer for this suggestion. As we detail in our response to e.g. Reviewer #1, Comment 3 we have now consistently added robust linear model fitting complemented by correcting for multiple hypothesis testing using FDR correction across all analyses in the manuscript. Supplementary Tables now include beta coefficients, 95% CIs, p-values and FDR corrected p-values and should thus now give a better impression of the magnitude of effect as well as the statistical confidence of our results. Where appropriate these data have also been added to revised Figures and Tables and are being discussed in the text.

4. The RNAi experiments are important here to address pairwise genetic interactions among the 30 validated causal genes. And indeed, some significant genetic interactions were found. However, it’s not clear to me how the scores were determined and why do they sometimes switch signs? (Figure 3c)

We thank the Reviewer for pointing out potential confusion that may arise from how the coRNAi scores were determined. We have carefully rephrased the respective Methods section and hope that it will be clearer now (pages 31-32).

In brief, for coRNAi we first determined robust Z-scores using the following formula:

$$\text{robust Z - score} = \frac{I_{F\text{treated}} - \text{median}I_{F\text{controls}}}{\text{mad}I_{F\text{controls}}}$$

in which $I_{F\text{treated}}$ represents the mean of the total fluorescence intensity of the Dil-LDL signal (LDL that was endocytosed) above the local background of all cells per image calculated per biological replica. A robust Z-score value was calculated for each treatment, namely for each single siRNA ([siRNA_{geneA}+Neg9] and [siRNA_{geneB}+Neg9]) or double siRNA knock-down ([siRNA_{geneA}+ siRNA_{geneB}]) where Neg9 is the negative control siRNA. The $\text{median}I_{F\text{controls}}$ is the median of total fluorescence signal of all the negative control siRNAs per array and the $\text{mad}I_{F\text{controls}}$ is the median absolute deviation of these controls. A median robust Z-score was calculated per treatment across all biological replicates for each gene pair. This median robust Z-score is reflected as heatmap color in Figure 3b for the primary screen, and as bar graph in Figure 3c for the validation coRNAi analysis.

Subsequently, we tested if simultaneous knockdowns result in either *additive effects* (AEs) or *genetic interactions* (GIs) on LDLc uptake by conducting Robust Linear Model fitting in R. We took robust Z-score values calculated from different biological replicates in the presence of single ([siRNA_{geneA}+Neg9] and [siRNA_{geneB}+Neg9]) or double knockdowns ([siRNA_{geneA}+ siRNA_{geneB}]) and considered them to be response variable values. Negative control values [Neg9] were included in each fitted dataset to correctly account for baseline LDLc uptake. The full regression model considered in the study was:

$$y = \beta_0 + \beta_A * X_A + \beta_B * X_B + \beta_{AB} * X_A * X_B + \epsilon$$

In Figure 3c, we represent in addition to the *median robust Z-score* (left panel) also the *Interaction Value* (right panel). This *Interaction Value* is denoted as β_{AB} in the above equation. *Interaction value* or interaction effect represents the difference between the observed robust Z-score values in case of double knockdown y_{AB} and the expected additive effect of geneA and geneB knockdown ($\beta_{AB} = y_{AB} - \beta_0 - \beta_A - \beta_B$). Thus, Interaction Values do not have to follow the directionality of the robust Z-score value for the double effect.

We exemplify this phenomenon at the case of the following two gene pairs, which our coRNAi analyses proposed as *positive alleviating GIs* (former nomenclature *buffering naGIs*), depicted here as Reviewer Figure 1.

Reviewer Figure 1

As the Reviewer can appreciate from this Figure, in both cases robust Z-score values were negative for double knockdowns, indicating decreased LDLc uptake upon joint knockdown of both genes of the

respective gene pair. However, the *Interaction Value* is negative only for *LDLR-LDLRAP1*, while for *SORT1-TOMM40* it is positive. This is because the difference between expected versus observed effect was in the first case *reduced* and in the latter *increased*.

For both gene pairs, one of the genes exerts a dominant role on the respective *positive alleviating* subtype GI. For the left gene pair, *LDLRAP1* cannot fully compensate the strong reduction in LDLc uptake upon knockdown of *LDLR*, whereas for the right example, *SORT1* knockdown does not further aggravate the strong LDLc uptake inhibition upon *TOMM40* knockdown.

In our revised Methods, we hope to provide now more clarity on how the coRNAi scores were determined and why they sometimes switch signs.

5. This manuscript does not really expand our knowledge concerning the genetic architecture of complex traits. However, it is likely relevant to researchers interested in lipid metabolism and CAD.

We did not claim that our findings would substantially expand our knowledge concerning the architecture of complex traits - apart from providing additional support for the hypothesis AEs massively outweigh GIs, at least for lipid traits and the tested gene set, and by now expanding this assumption to high-impact PTVs and coRNAi. To avoid a risk of overstatement, our revised manuscript now abstains from any reference to “genetic architecture”.

REVIEWERS' COMMENTS

Reviewer #1 (Remarks to the Author):

General comments

The revised manuscript by Zimon et al. (now entitled "Pairwise effects between lipid GWAS genes modulate lipid plasma levels and cellular uptake") remains of high interest as it investigates the nature and extent of interaction and additivity for genes that are important in the context of heart disease. The authors now more clearly argue that harnessing not only synergistic genetic interactions but also additivity can open new avenues to combinatorial drug treatments aimed at targeting a particular disease or phenotype. To do so, they identify GWAS loci implicated in lipid phenotypes and harness three parallel types of information to uncover not only positive and negative genetic interactions, but also genetic interactions of additive effects, which they rightly claim to be of high interest when considering new therapeutic avenues.

The authors have done a great job of addressing previous concerns and comments in their updated manuscript. There are no remaining major concerns, only minor issues listed below.

Minor issues

1. Response to review item 15 ('Lines 68-69: Should cite substantial earlier literature that GIs can assist in nominating drug pair therapeutics.'): The addition of more references here was good, but authors may want to reconsider whether refs 15 and 16 are needed. Although Ref 15 offers some evidence that GIs can assist in nominating drug pair therapeutics, perhaps its major point is that only using GIs would miss many synergistic drug combinations. The focus of Ref 16 is on genetic interactions between null and over-expression alleles.

2. Lines 229-234: The authors may wish to more explicitly mention how surprising is the finding that knockdown of LDLRAP1 leads to increase LDLc uptake, highlighting more clearly that this is contrary to our current understanding of how this complex works. This could perhaps be flagged as an area of future study.

Reviewer #2 (Remarks to the Author):

Thank you for the opportunity to re-review this substantially revised manuscript. I have no major persistent concerns. Overall, this will be a nice contribution to the field.

Reviewer #3 (Remarks to the Author):

The Reviewers have done an excellent job of addressing my concerns and, as far as I can tell, those of the other Reviewers. In my opinion, this manuscript is now in very good shape!

Response to the Reviewers comments:

We thank all the Reviewers for their positive comments on our manuscript and their help in further improving it.

Reviewer #1 (Remarks to the Author):

General comments

The revised manuscript by Zimon et al. (now entitled “Pairwise effects between lipid GWAS genes modulate lipid plasma levels and cellular uptake”) remains of high interest as it investigates the nature and extent of interaction and additivity for genes that are important in the context of heart disease. The authors now more clearly argue that harnessing not only synergistic genetic interactions but also additivity can open new avenues to combinatorial drug treatments aimed at targeting a particular disease or phenotype. To do so, they identify GWAS loci implicated in lipid phenotypes and harness three parallel types of information to uncover not only positive and negative genetic interactions, but also genetic interactions of additive effects, which they rightly claim to be of high interest when considering new therapeutic avenues.

The authors have done a great job of addressing previous concerns and comments in their updated manuscript. There are no remaining major concerns, only minor issues listed below.

Minor issues

1. Response to review item 15 (‘Lines 68-69: Should cite substantial earlier literature that GIs can assist in nominating drug pair therapeutics.’): The addition of more references here was good, but authors may want to reconsider whether refs 15 and 16 are needed. Although Ref 15 offers some evidence that GIs can assist in nominating drug pair therapeutics, perhaps its major point is that only using GIs would miss many synergistic drug combinations. The focus of Ref 16 is on genetic interactions between null and over-expression alleles.

Following Reviewer #1’s recommendation we have removed the two respective references.

2. Lines 229-234: The authors may wish to more explicitly mention how surprising is the finding that knockdown of LDLRAP1 leads to increase LDLc uptake, highlighting more clearly that this is contrary to our current understanding of how this complex works. This could perhaps be flagged as an area of future study.

Following the reviewer’s suggestion, the respective sentence in the manuscript now reads:

Conversely, knockdown of LDLR strongly inhibited, whereas knockdown of LDLRAP1 increased cellular LDLc uptake under our experimental conditions, an effect that could not be explained by elevated LDLR levels at the plasma membrane (Supplementary Fig. 5) or would have been anticipated from LDLRAP’s role in FH^(ref.).

Reference:

Garcia CK, Wilund K, Arca M, Zuliani G, Fellin R, Maioli M, Calandra S, Bertolini S, Cossu F, Grishin N, Barnes R, Cohen JC, Hobbs HH. Autosomal recessive hypercholesterolemia caused by mutations in a putative LDL receptor adaptor protein. Science. 2001 May 18;292(5520):1394-8.

Reviewer #2:

Thank you for the opportunity to re-review this substantially revised manuscript. I have no major persistent concerns. Overall, this will be a nice contribution to the field.

We thank the reviewer for their appreciation of and help in shaping our study.

Reviewer #3:

The Reviewers have done an excellent job of addressing my concerns and, as far as I can tell, those of the other Reviewers. In my opinion, this manuscript is now in very good shape!

We thank the reviewer for their appreciation of and help in shaping our study.